# Cross-modal interaction of human alpha activity does not reflect inhibition of early sensory processing in a frequency-tagging study using EEG and MEG

Marion Brickwedde[1,2,3]*, Rupali Limachya[3], Roksana Markiewicz[3,4], Emma Sutton[3,5], Christopher Postzich[6], Kimron Shapiro[3], Ole Jensen[3,7], Ali Mazaheri[3]

[1]Department of Child and Adolescent Psychiatry, Charité - Universitätsmedizin Berlin, Berlin, Germany; [2]Physikalisch Technische Bundesanstalt, Berlin, Germany; [3]Centre for Human Brain Health, School of Psychology, University of Birmingham, Birmingham, United Kingdom; [4]Istituto Italiano di Tecnologia, Genova, Italy; [5]Bristol Trials Centre, Bristol Medical School, University of Bristol, Bristol, United Kingdom; [6]Max-Planck Research Institute, Leipzig, Germany; [7]Department for Translational Cognitive Neuroscience, University of Oxford, Oxford, United Kingdom

*For correspondence:
marion.brickwedde@charite.de

Competing interest: The authors declare that no competing interests exist.

## eLife Assessment

This **valuable** manuscript provides **solid** evidence regarding the role of alpha oscillations in sensory gain control. The authors use an attention-cuing task in an initial EEG study followed by a separate MEG replication study to demonstrate that whilst (occipital) alpha oscillations are increased when anticipating an auditory target, so is visual responsiveness as assessed with frequency tagging. The findings offer a re-interpretation of the inhibitory role of the alpha rhythm, supporting that alpha oscillations contribute to interareal communication.

**Abstract** Selective attention involves prioritising relevant sensory input while suppressing irrelevant stimuli. It has been proposed that oscillatory alpha-band activity (~10 Hz) aids this process by functionally inhibiting early sensory regions. However, recent studies have challenged this notion. Our EEG and MEG studies aimed to investigate whether human alpha oscillations serve as a 'gatekeeper' for downstream signal transmission. We first observed these effects in an EEG study and then replicated them using MEG, which allowed us to localise the sources. We employed a cross-modal paradigm where visual cues indicated whether upcoming targets required visual or auditory discrimination. To assess inhibition, we utilised frequency-tagging, simultaneously flickering the fixation cross at 36 Hz and playing amplitude-modulated white noise at 40 Hz during the cue-to-target interval. Consistent with prior research, we observed an increase in posterior alpha activity following cues signalling auditory targets. However, remarkably, both visual and auditory frequency-tagged responses amplified in anticipation of auditory targets, correlating with alpha activity amplitude. Our findings suggest that when attention shifts to auditory processing, the visual stream remains responsive and is not hindered by occipital alpha activity. This implies that alpha modulation does not solely regulate 'gain control' in early visual areas but rather orchestrates signal transmission to later stages of the processing stream.

## Introduction

In our daily life, we are often confronted with sensory information from many different sources, all at once. To operate effectively, we require selective attention, reconciling the tension between environmental inputs relevant for top-down goals and sensory information that may be perceptually salient but task-irrelevant. Previous research gave rise to the prominent alpha inhibition hypothesis, which suggests that oscillatory activity in the alpha range (~10 Hz) plays a mechanistic role in selective attention through functional inhibition of irrelevant cortical areas (see *Figure 1*; *Foxe et al., 1998*; *Jensen and Mazaheri, 2010*; *Klimesch et al., 2007*). Functional inhibition refers to an area of the cortex being actively hindered to process input, which is distinctly different from idling, where a part of the cortex is not actively involved. Evidence supporting this theory revealed an increase in alpha-power over task-irrelevant sensory cortices after the onset of cues indicating the spatial location (*Kelly et al., 2006*; *Okazaki et al., 2014*; *Thut et al., 2006*; *Worden et al., 2000*; *Zumer et al., 2014*) or specific modality of an upcoming target (*Foxe et al., 1998*; *Fu et al., 2001*; *Mazaheri et al., 2014*). Moreover, previous investigations have observed 'spontaneous fluctuations' of alpha power in sensory regions, particularly in the visual cortex, to be inversely related to discrimination ability (*Ergenoglu et al., 2004*; *van Dijk et al., 2008*). Alpha inhibition is believed to be transmitted in a phasic manner, as phosphene perception as well as high-frequency and spiking activity vary in line with the alpha cycle (*Dugué et al., 2011*; *Haegens et al., 2011*; *Spaak et al., 2012*).

Recent evidence challenged a direct connection between alpha activity and visual information processing in early visual cortex. As such, both visual steady-state responses and alpha power were modulated by attention but did not covary when investigating individual trials (*Zhigalov and Jensen, 2020*). Unfortunately, very few studies have investigated direct connections between alpha activity, attention and sensory signals, especially over trials. Furthermore, results seem to depend on timing of alpha activity in relation to sensory responses as well as stimulus type and outcome measure (*Morrow et al., 2023*).

Accordingly, the objective of the current study is to test the alpha inhibition hypothesis compared to an alternative theory. Based on the alpha inhibition hypothesis, alpha modulation is connected to 'gain control' in early visual areas through modulation of excitability (*Foxe and Snyder, 2011*; *Jensen and Mazaheri, 2010*; *Van Diepen et al., 2019*). In contrast, we propose that functional and inhibitory effects of alpha modulation, such as distractor inhibition, are exhibited through blocking or facilitating signal transmission to higher order areas (*Peylo et al., 2021*; *Yang et al., 2023*; *Zhigalov and Jensen, 2020*; *Zumer et al., 2014*), gating feedforward or feedback communication between sensory areas (see *Figure 1*; *Bauer et al., 2020*; *Haegens et al., 2015*; *Uemura et al., 2021*).

To this end, we applied frequency-tagging, the rhythmic presentation of sensory stimuli, which elicits steady-state sensory evoked potentials or fields (SSEP/SSEF), consisting of rhythmic neuronal activity in the frequency of stimulation (*Brickwedde et al., 2020*; *Colon et al., 2012*; *Dinse et al., 2003*; *Marzoll et al., 2018*; *Regan, 1982*; *Snyder, 1992*; *Stapells et al., 1984*; *Tobimatsu et al., 1999*). The magnitude of SSEPs is attention-dependent (*de Jong et al., 2010*; *Müller et al., 1998*; *Müller and Hillyard, 2000*; *Porcu et al., 2013*; *Saupe et al., 2009*; *Toffanin et al., 2009*) even for frequencies too fast to perceive consciously (*Brickwedde et al., 2022*; *Zhigalov et al., 2019*). Visual SSEP/SSEFs are most strongly observable over occipital areas, whereas auditory SSEP/SSEFs appear most strongly over temporal (MEG) or fronto-to-central (EEG) areas (*de Jong et al., 2010*; *Hari et al., 1989*; *Pantev et al., 1996*; *Regan, 1982*). Furthermore, when applying two different frequencies for two different sensory modalities, their intermodulation frequency (f1-f2) has been suggested to reflect cross-modal integration (*Drijvers et al., 2021*). Due to distinct responses, localisation and attention-dependence, frequency-tagging provides an optimal tool to study sensory signal processing and integration over time.

The aim of our study was to directly test the alpha inhibition hypothesis by investigating if cue-induced modulation of alpha activity coincides with the suppression of frequency-tagging responses in task-irrelevant modalities. Based on previous studies, we utilised a cross-modal attention paradigm, in which symbolic visual cues signalled the target modality (visual or auditory) of an upcoming discrimination task (e.g. *van Diepen et al., 2015*). Here, we included an additional experimental manipulation in the form of frequency-tagging to assess the involvement of the auditory and visual systems in the *cue-to-target interval*.

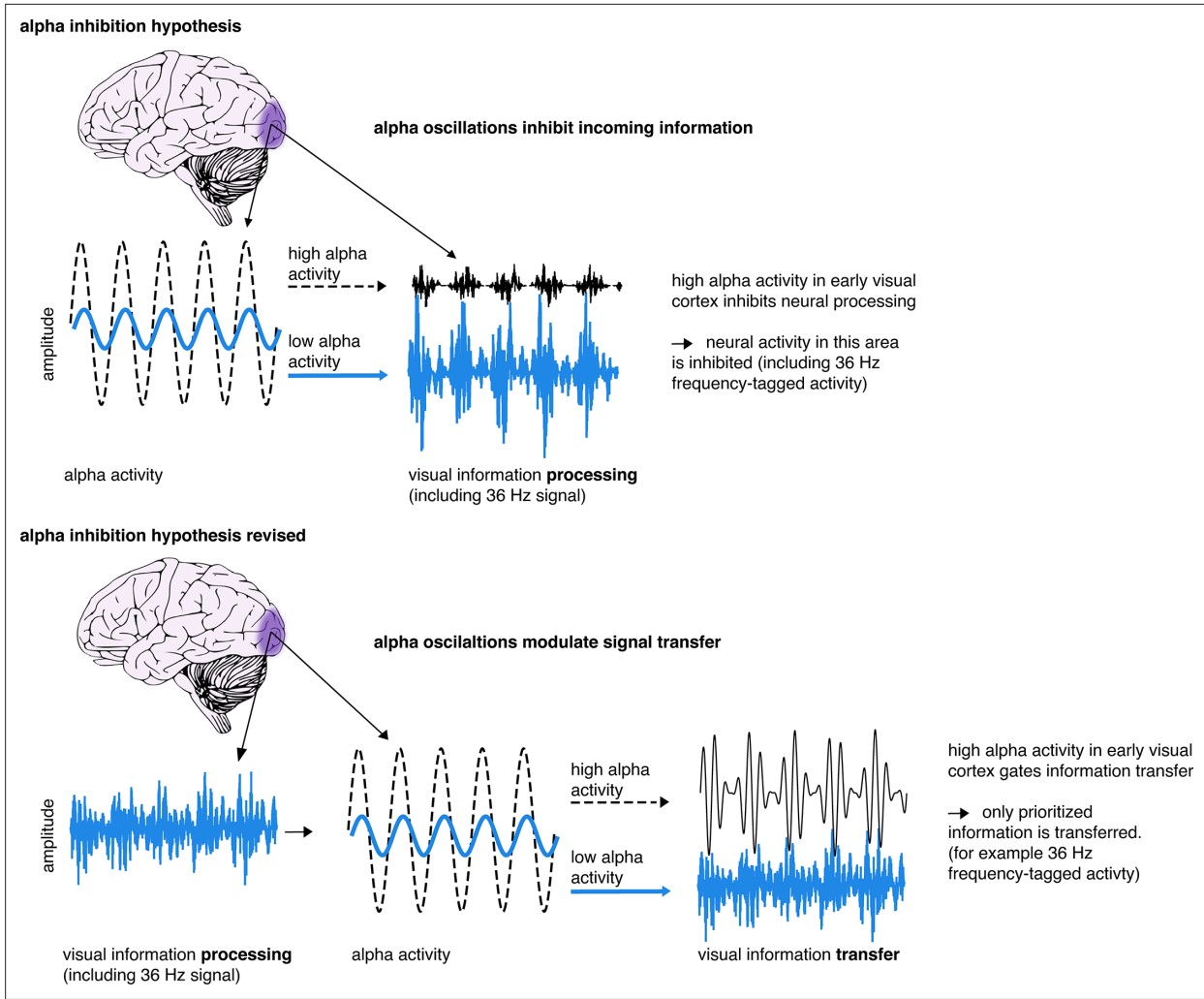

**Figure 1.** Illustration of the alpha inhibition hypothesis. The alpha inhibition hypothesis suggests that occipital alpha inhibits visual information processing in a phasic manner. If alpha activity is high, it suggests that neural processing in the cortical area is inhibited (*Foxe et al., 1998*; *Jensen and Mazaheri, 2010*; *Klimesch et al., 2007*). We propose a revision of this hypothesis, whereby alpha activity exerts its phasic inhibition to regulate information transfer, creating enhanced signal packages of prioritised information (see also *Yang et al., 2023*; *Zhigalov and Jensen, 2020*; *Zumer et al., 2014*). This way, irrelevant or distracting information is inhibited through a block of transfer, rather than through blocking of incoming sensory information.

In line with previous results, we hypothesised that signalling an upcoming auditory target would lead to increased alpha activity over visual occipital regions as well as increased SSEP responses to auditory and decreased SSEP to visual stimuli as indexed by frequency-tagging. In brief, while we observed the expected cue-induced early-visual alpha modulation, the amplitude of auditory and *visual* SSEP/SSEFs as well as their intermodulation frequency increased just prior to the onset of the auditory target, contradicting the alpha inhibition hypothesis. The difference between conditions of visual SSEP/SSEFs originated from sensory integration areas and correlated with early sensory alpha activity on a trial-by-trial basis, speaking to an effect of alpha modulation on signal transmission rather than inhibition of early visual areas.

## Results
### Study 1: Cross-modal EEG experiment
To assess audio-visual excitability in anticipation of either visual or auditory targets, we displayed visual cues to signal the modality of the upcoming target (auditory, visual, or non-specific). In a three-second

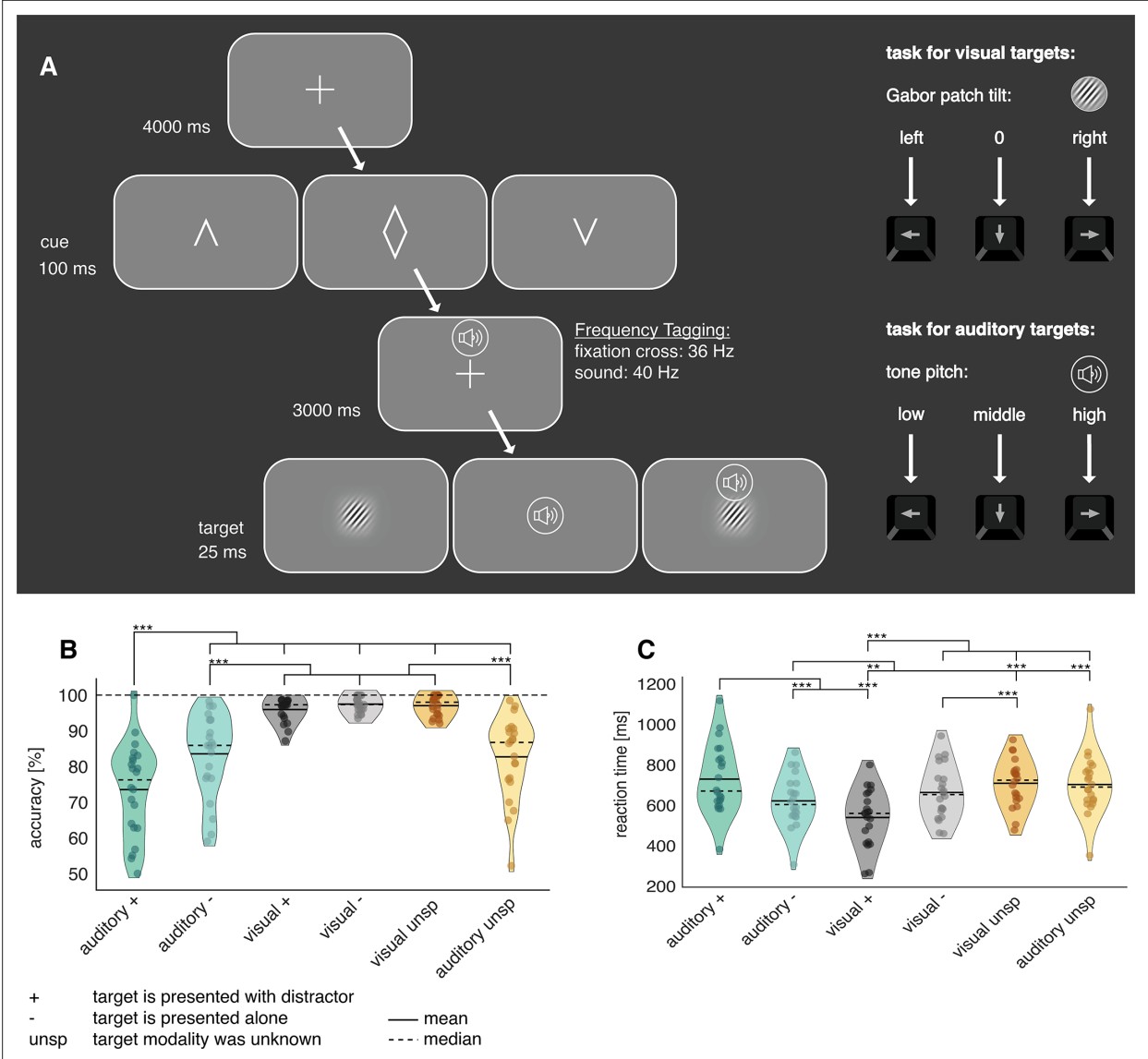

**Figure 2.** Illustration of the cross-model discrimination task in study 1. (**A**) Trials were separated by a 4 s interval, in which a fixation cross was displayed. A brief central presentation of the cue (100 ms) initiated the trial, signalling the target modality (see figure above from left to right: auditory, non-specific, visual). In the cue-to-target interval, the fixation cross was frequency-tagged at 36 Hz. At the same time, a sound was displayed over headphones, which was frequency-tagged at 40 Hz. Both tones and fixation cross contained no task-relevant information. The target, consisting either of a static Gabor patch or a tone, was presented for 25ms. Participants had to differentiate via button presses between three different Gabor rotations or tone pitches, respectively. In 50% of auditory and visually cued trials, a distractor in form of a random pitch or rotation of the un-cued modality was presented alongside the target. (**B, C**) Analysis of task accuracy and reaction time indicates increased difficulty of auditory targets. (**B**) Task accuracy compared between all six experimental conditions reveals a drop in accuracy for responses to auditory targets. (**C**) Reaction times of correct trials compared between all six experimental conditions. The slowest reaction times are observable following auditory targets alongside visual distractors. There was an attentional benefit of cues for reaction times and a distractor cost for accuracy and reaction times (see *Figure 2—figure supplement 1*) N=22; *** sig <0.001; ** sig.<0.01; * sig.<0.05.

The online version of this article includes the following figure supplement(s) for figure 2:

**Figure supplement 1.** Distractor cost and attentional benefit.

cue-to-target interval, we frequency-tagged the fixation cross at 36 Hz and played 40 Hz amplitude modulating white noise. Immediately afterwards, participants had to either discriminate between three different pitch sounds (auditory target) or three different Gabor patch orientations (visual target). If the target modality was cued (e.g. not non-specific), 50% of the trials were accompanied by a random distractor from the target pool of the opposing sensory modality (see *Figure 2A*).

## Behavioural performance

We found that accuracy differed significantly between conditions ($F_{(5,105)}$ = 44.16; p<0.001). Overall, participants were significantly less accurate in the auditory discrimination task ('overall auditory', M=79% correct, SD = 11.7) than in the visual discrimination task ('overall visual', M=97%, SD = 2.3; see *Figure 2B*), with the worst performance occurring when auditory targets were paired with visual distractors (auditory +: M=74% correct, SD = 13).

Reaction times yielded a similar pattern, with auditory reactions (overall auditory: M=662ms, SD = 136) being slower than for the visual task (overall visual: M=597 ms, SD = 130; main effect over all conditions: $F_{(5,105)}$ = 27.47; p<0.001; see *Figure 2C*). This was mostly driven by slow responses to auditory targets paired with distractors (auditory +: M=723 ms).

To ascertain that cues and distractors were functionally relevant, we calculated attentional benefit (cued – non-specific condition) and distractor cost (cued condition with distractor – cued condition without distractor). As expected, cues improved reaction times over all conditions and distractors impaired the accuracy over all conditions and reaction times for auditory targets. (see *Figure 2— figure supplement 1*). Interestingly, auditory distractors reduced reaction times to visual targets, which could be explained by a generally faster processing of auditory targets (*Jain et al., 2015*). As such, the auditory distractor possibly caused intersensory facilitation (*Nickerson, 1973*), whereby

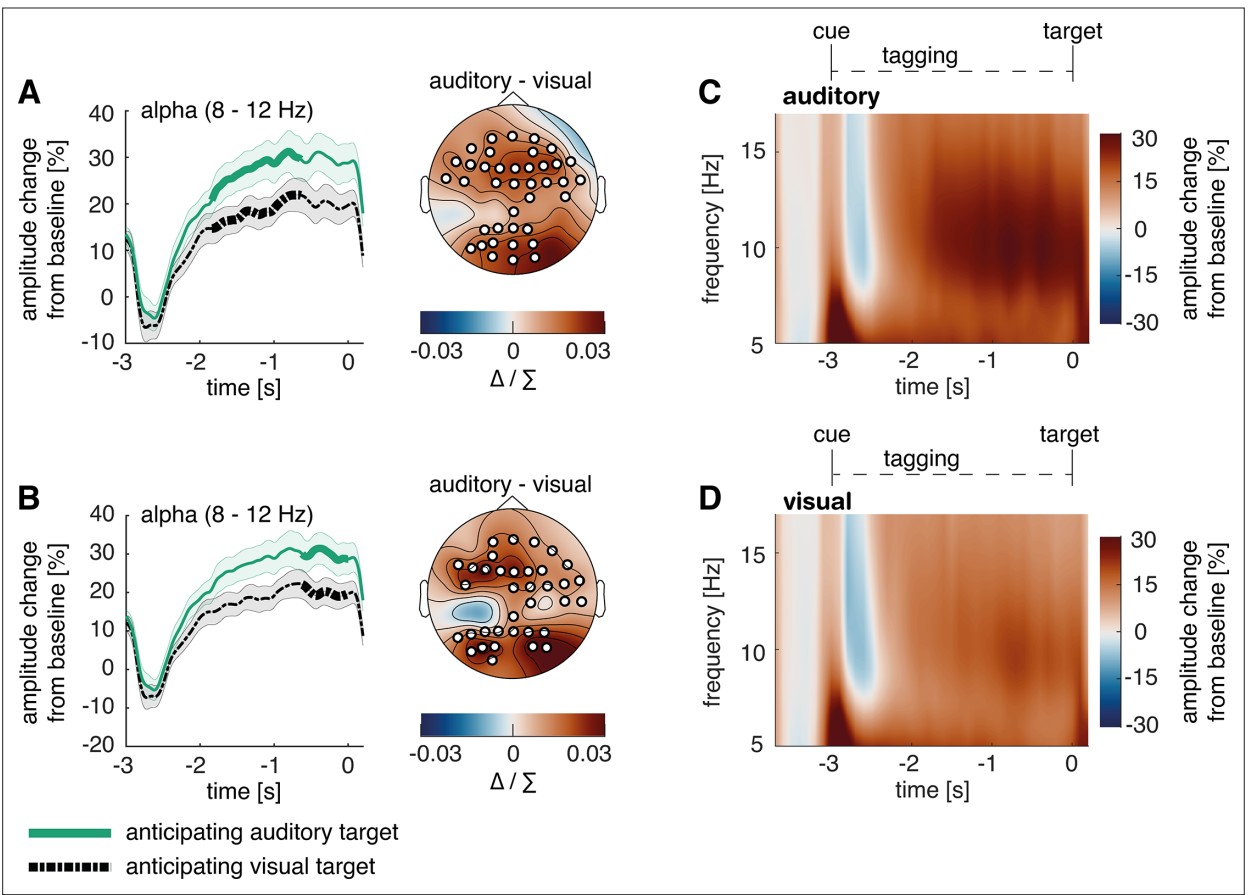

**Figure 3.** Post-cue modality-specific modulation of alpha power in anticipation of an auditory versus a visual target. (**A, B**) The time course of post-cue alpha power. Cluster permutation analysis resulted in two condition effects, both indicating heightened alpha activity when expecting an auditory compared to a visual target (**A**: p<0.01; **B**: p<0.01). For a comparison to trials with non-specific cues, see *Figure 3—figure supplement 1*. (**C, D**) Time-frequency representation of power in the cue-to-target interval. A greater increase in alpha power was observed when expecting an auditory target (average over significant electrodes for the condition difference in **A**). *Note:* Cluster electrodes are marked in white. Shading represents standard error from the mean; Δ / Σ represents (a-b)/(a+b) normalisation.

The online version of this article includes the following figure supplement(s) for figure 3:

**Figure supplement 1.** Time course of alpha activity and frequency-tagging responses for the non-specific compared to the visually-cued condition.

reaction times to a target can be facilitated when accompanied by stimuli of other sensory modalities, even if they are irrelevant or distracting.

## Cross-modal cues differentially modulated pre-target alpha activity

We conducted a time-frequency analysis of power in the cue-to-target interval and found a stronger amplitude increase from baseline for auditory compared to visual target conditions starting around 2 s before target onset (*Figure 3*). We calculated the time course of alpha power changes using the Hilbert-transformation (8–12 Hz, *Figure 3A and B*). We then applied cluster permutation analysis, whereby real condition differences were tested against coincidental findings by randomly permuting the condition labels to the data and testing for condition differences 1000 times (*Maris and Oostenveld, 2007*). Consistent with previous work (*Mazaheri et al., 2014*; *van Diepen and Mazaheri, 2017*), analysing the last two seconds before target onset revealed two significant clusters of difference in alpha power when expecting an auditory compared to a visual target. These effects corresponded to clusters extending from –1.84 to – 0.64 s (p=0.004) and –0.62–0 s (p=0.005). When presenting non-specific cues, alpha power changes were not significant, but descriptively larger compared to visual target conditions and lower compared to auditory target conditions (see *Figure 2—figure supplement 1*). However as significant alpha modulation was a prerequisite to test our hypotheses, we excluded this condition from further analysis.

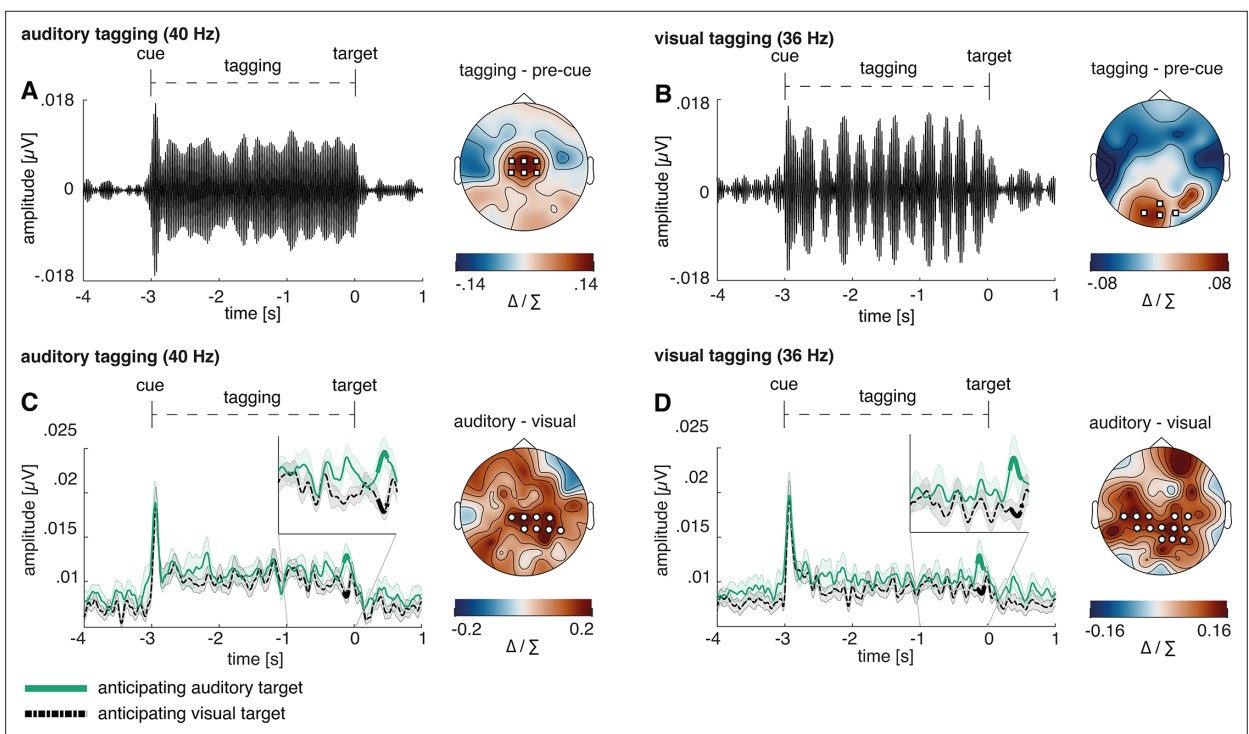

**Figure 4.** Increase in amplitude of both visual and auditory frequency-tagged responses when anticipating visual or auditory targets. Event-related potentials and scalp topographies reveal distinct modality-specific responses at the tagged frequencies. (**A**) Auditory steady-state evoked potential (ASSEP) averaged over six central electrodes displaying the highest 40 Hz power (Cz, FC1, FC2, C1, C2, FCz; marked with white squares). (**B**) Visual steady-state evoked potential (VSSEP) averaged over four occipital electrodes displaying the highest 36 Hz power (POz, O1, O2, Oz; marked with white squares). (**C**) The Hilbert envelope of the 40 Hz ASSEP reveals an increase shortly before target onset when anticipating an auditory compared to a visual target (p=0.041); (**D**) the Hilbert envelope of the 36 Hz VSSEP likewise reveals an increase shortly before target onset when anticipating an auditory compared to a visual target (p=0.014). For individual power spectra across conditions, see *Figure 4—figure supplements 1 and 2*. For a comparison to trials with non-specific cues, see *Figure 3—figure supplement 1*. *Note*: Cluster electrodes are marked in white. Shading represents standard error from the mean. Δ / Σ represents (a-b)/(a+b) normalisation.

The online version of this article includes the following figure supplement(s) for figure 4:

**Figure supplement 1.** Individual ERP power spectra of the cue-to-target interval when anticipating an auditory target in the EEG-study.

**Figure supplement 2.** Individual ERP power spectra of the cue-to-target interval when anticipating a visual target in the EEG-study.

## Cross-modal cues increased the amplitude of the frequency-tagged responses across both modalities

To assess the temporal development of frequency-tagging responses, steady-state potentials were calculated using data band-pass filtered around the tagging frequency. Neuronal 40 Hz auditory-steady state evoked potential (ASSEP) responses were strongest over central areas (*Figure 4A*), and 36 Hz auditory-steady state evoked potential (VSSEP) responses were strongest over occipital areas (*Figure 4B*).

To assess the differences between conditions, the Hilbert envelope of the steady-state potentials was analysed using cluster permutation analyses. When expecting an auditory target compared to a visual target, the 40 Hz ASSEP response was larger shortly before target onset (see *Figure 4C*; –0.15 to –0.08 s; p=0.041).

Surprisingly, the 36 Hz VSSEP response was likewise increased shortly before expecting an auditory compared to a visual target (see *Figure 4D*; –0.16 to –0.06 s; p=0.014). As such, the observed frequency-tagging responses might reflect effort, which affects the vigilance of the whole sensory system rather than sensory-specific allocations of attention. For both 36 Hz VSSEP and 40 Hz ASSEP

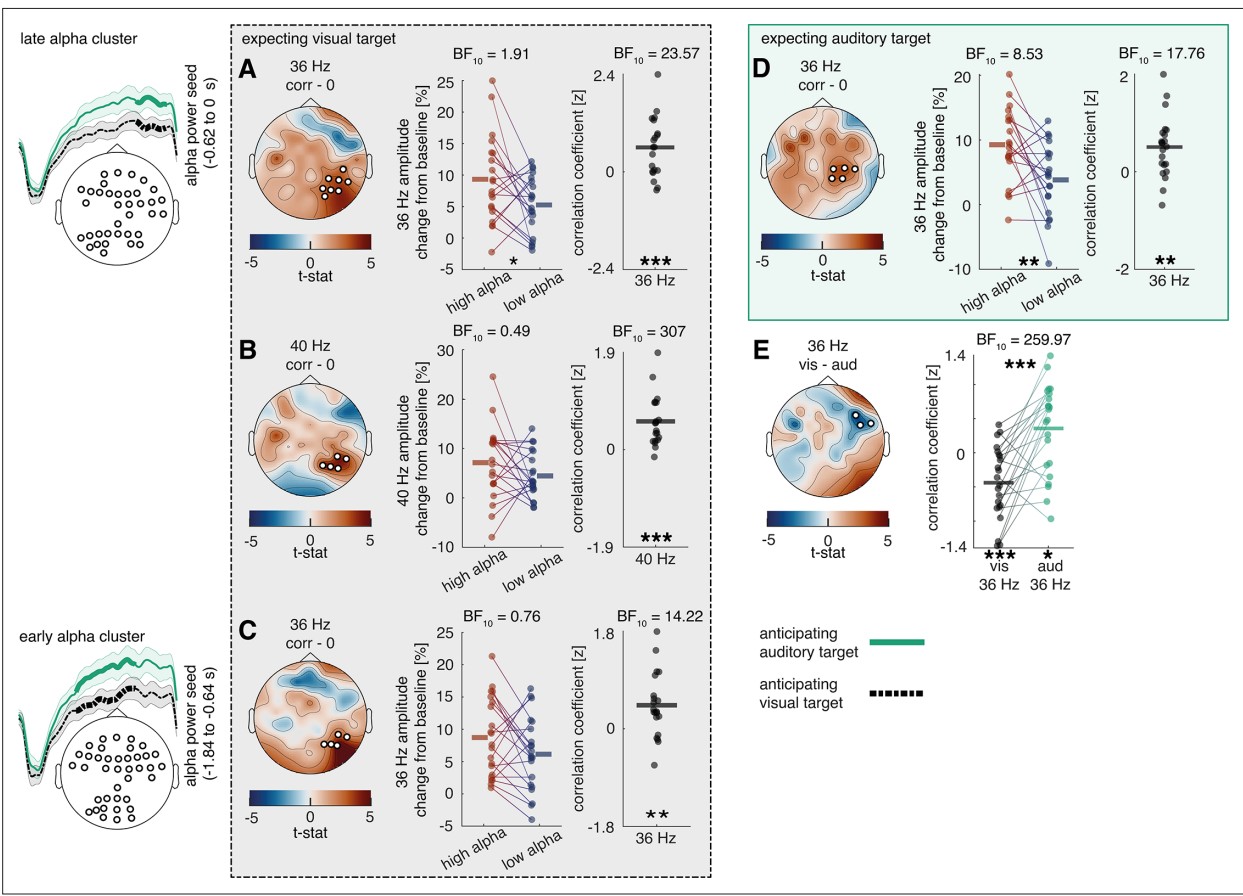

**Figure 5.** Relationship between cue-induced alpha modulation and amplitude of frequency-tagged responses in study 1. Previously obtained alpha clusters (see *Figure 3*) were correlated over trials with auditory 40 Hz and visual 36 Hz clusters (see *Figure 4*), where alpha electrodes/sensors were applied as seeds. The analysis was performed using a cluster-permutation approach, testing a correlation model against a 0-correlation model. Clusters significantly diverging from the 0-correlation model are presented topographically. Additionally, median splits between high and low alpha trials as well as correlation coefficients of these clusters are displayed for all participants. (**A, B**) A positive correlation is visible between alpha activity in the last 400 ms and steady state potentials shortly before target onset when expecting a visual target (visual 36 HZ: p=0.013; 40 Hz: p=0.009). (**D**) When expecting an auditory target, there is a positive correlation between alpha activity in the last 400 ms and visual 36 Hz activity shortly before target onset (p=0.010). (**E**) The correlation between alpha activity 400 ms and visual 36 Hz activity shortly before target onset changes its direction depending on whether an auditory or a visual target is expected (p=0.037). (**C**) A positive correlation is also visible between alpha activity as early as ~1200 ms to 400 ms and visual 36 Hz activity shortly before target onset when expecting a visual target (p=0.016). N=22; *** sig <0.001; ** sig <0.01; * sig <0.05.+sig.<0.1.

responses, condition differences appeared strongest over mid-parietal regions in contrast to primary sensory cortices.

## Alpha power was positively correlated with amplitude of frequency-tagged responses

Following the observation of condition differences in alpha activity and frequency-tagging responses, we were further interested in exploring whether these responses were connected. Accordingly, we conducted trial-by-trial correlations using alpha condition differences and their electrode positions as seed, which was correlated with frequency-tagging signals over all electrodes. Multiple comparison correction was applied by testing the correlation matrix against a zero-correlation matrix with a cluster permutation approach. A positive correlation was observed over right parietal-to-occipital areas between the late alpha cluster activity and both 40 Hz ASSEP (p=0.009) and 36 Hz VSSEP (p=0.004) responses when expecting a visual target (see *Figure 5A and B*). To estimate the robustness of these results, we additionally conducted median split analyses between trials with high and low alpha power for each participant, as well as averaged the correlation coefficient of each participant and calculated a one-sample *t*-test against 0. For each analysis we provided the Bayes Factor, which estimates the strength of support for or against the null hypothesis (BF >3.2 is considered as substantial evidence and BF >10 is considered as strong evidence; *Kass and Raftery, 1995*).

The median split was highly significant for the 36 Hz VSSEP response (*Figure 5A*, middle columns, p=0.033; $t_{(19)}$ = 2.29; $BF_{(10)}$ = 1.91) but did not reach significance for the 40 Hz ASSEP response (*Figure 5B*, middle column; p=0.20; $t_{(19)}$ = 1.32; $BF_{(10)}$ = 0.49). Correlation coefficients indicated strong correlations with alpha activity for both 40 Hz ASSEP (*Figure 5B*, right column; p<0.001; $t_{(19)}$ = 4.95; $BF_{(10)}$ = 306.93) and 36 Hz VSSEP activity (*Figure 5A*, right column; p<0.01; $t_{(19)}$ = 3.66; $BF_{(10)}$ = 23.57).

The same positive correlation with alpha activity was found when expecting an auditory target for 36 Hz VSSEP activity (p=0.031), but not for 40 Hz ASSEP activity (see *Figure 5D*). A median split between high and low alpha activity (p=0.005; $t_{(19)}$ = 3.14; $BF_{(10)}$ = 8.53) and correlation coefficients (p=0.002; $t_{(19)}$ = 3.52; $BF_{(10)}$ = 17.76) provided moderate to strong evidence for this effect. The likely origin of alpha modulations from early visual cortices may explain a stronger connection to 36 Hz VSSPE responses compared to 40 Hz ASSEP responses.

Additionally, we compared how correlation coefficients between alpha activity and frequency-tagging differed when anticipating an auditory versus a visual target. Multiple comparison correction was applied with cluster permutation analysis. Interestingly, an interaction between the strength of the correlation associating alpha, 36 Hz VSSEP activity and condition became apparent (p=0.044; *see Figure 5E*) and was observed most strongly over right-central electrodes. Comparing the correlation coefficients of participants over this cluster revealed a strong effect and even a change of direction in the correlation (p<0.001; $t_{(21)}$ = –4.76; $BF_{(10)}$ = 259.97). Particularly, when expecting a visual target, there was a negative correlation between 36 Hz VSSEP and alpha activity, which turned positive when expecting an auditory target. Both correlations also differed significantly from 0 (expecting a visual target: p<0.001; $t_{(21)}$ = –3.87; $BF_{(10)}$ = 39.62; expecting an auditory target: p=0.020; $t_{(21)}$ = 2.53; $BF_{(10)}$ = 2.83). In contrast to the previously observed positive correlation between alpha activity and 36 Hz VSSEP activity, the significant electrode cluster was located more ventrally. This effect could possibly hint at dynamic adaptability of oscillatory alpha effects on later processing stages.

It is further noteworthy that the correlation between alpha activity and 36 Hz VSSEP response when expecting a visual target was also present when utilising the early alpha cluster as seed (~1200–400 ms before target onset, see *Figure 5C*), in which case alpha modulation greatly preceded the 36 Hz VSSEP modulation (p=0.016). For this correlation, the median split between high and low alpha trials did not reach significance (p=0.11; $t_{(20)}$ = 1.69; $BF_{(10)}$ = 0.76); however, testing the correlation coefficients against 0 again revealed a significant effect (p=0.003; $t_{(20)}$ = 3.39; $BF_{(10)}$ = 14.22). This result may support the directionality of alpha modulation affecting visual information processing (and not vice versa).

## Intermodulation frequency

Lastly, we analysed the steady-state response of the intermodulation frequency at 4 Hz. Increased intermodulation shortly before target onset could be observed when expecting an auditory compared to a visual target (−0.51 to −0.0620; p<0.001). This effect was strongest over left fronto-to-central

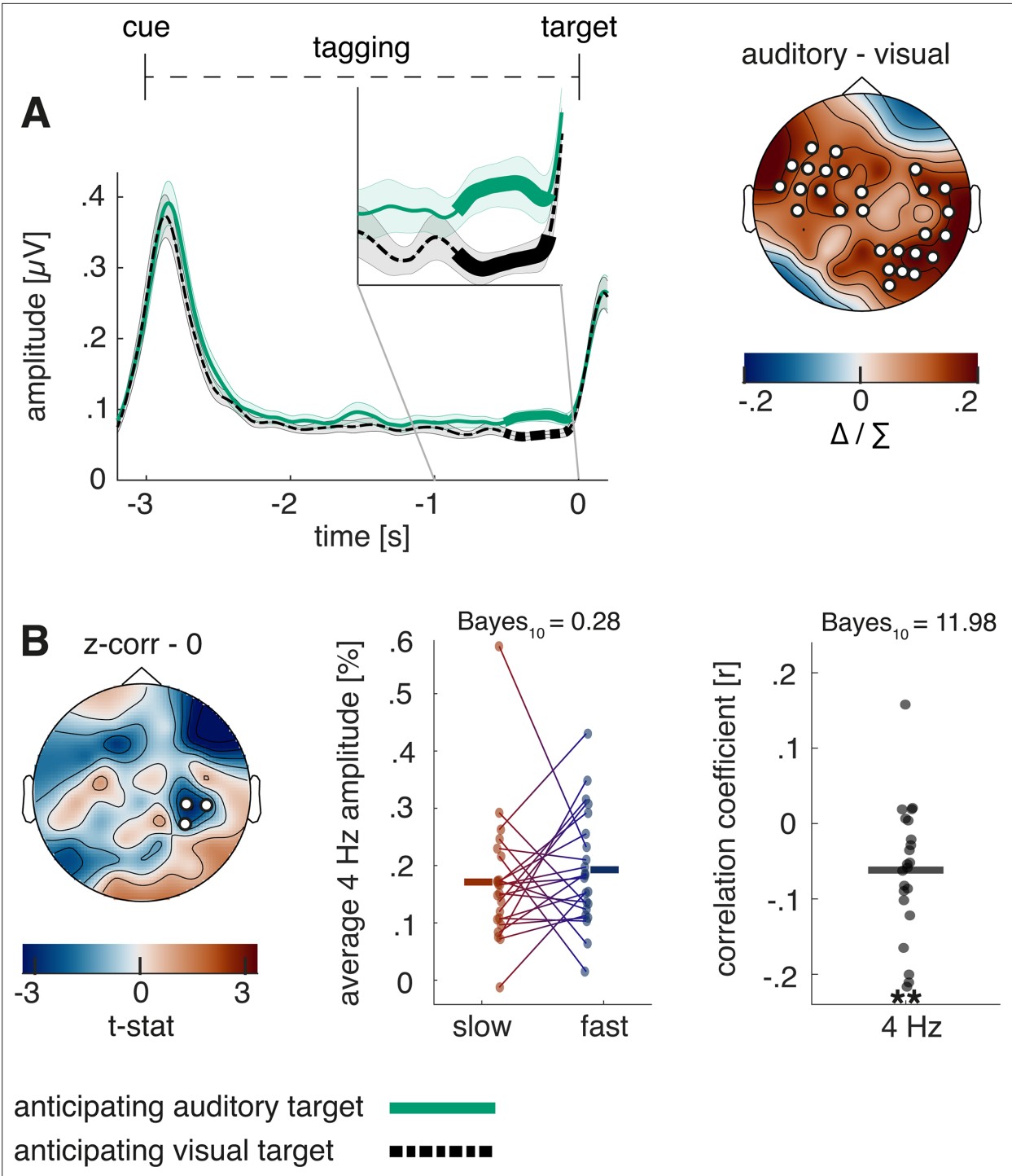

**Figure 6.** Steady-state response in the intermodulation frequency and its behavioural relevance. (**A**) The Hilbert envelope of the 4 Hz steady-state response reveals an increase shortly before target onset when anticipating an auditory compared to a visual target (p<0.01). (**B**) There is a trial-by-trial correlation between 4 Hz activity and reaction time when a visual target without distractor was presented. The correlation is further illustrated by a median split between fast and slow reaction time trials as well as by correlation coefficients for each participant. N=22; ** sig.<0.01.

electrodes and right central-to-occipital electrodes (see *Figure 6A*) and may suggest an increase in cross-modal integration.

With the goal to examine whether there are any behavioural effects of the condition difference, trial by trial correlations with reaction times of each of the two auditory and visual conditions were

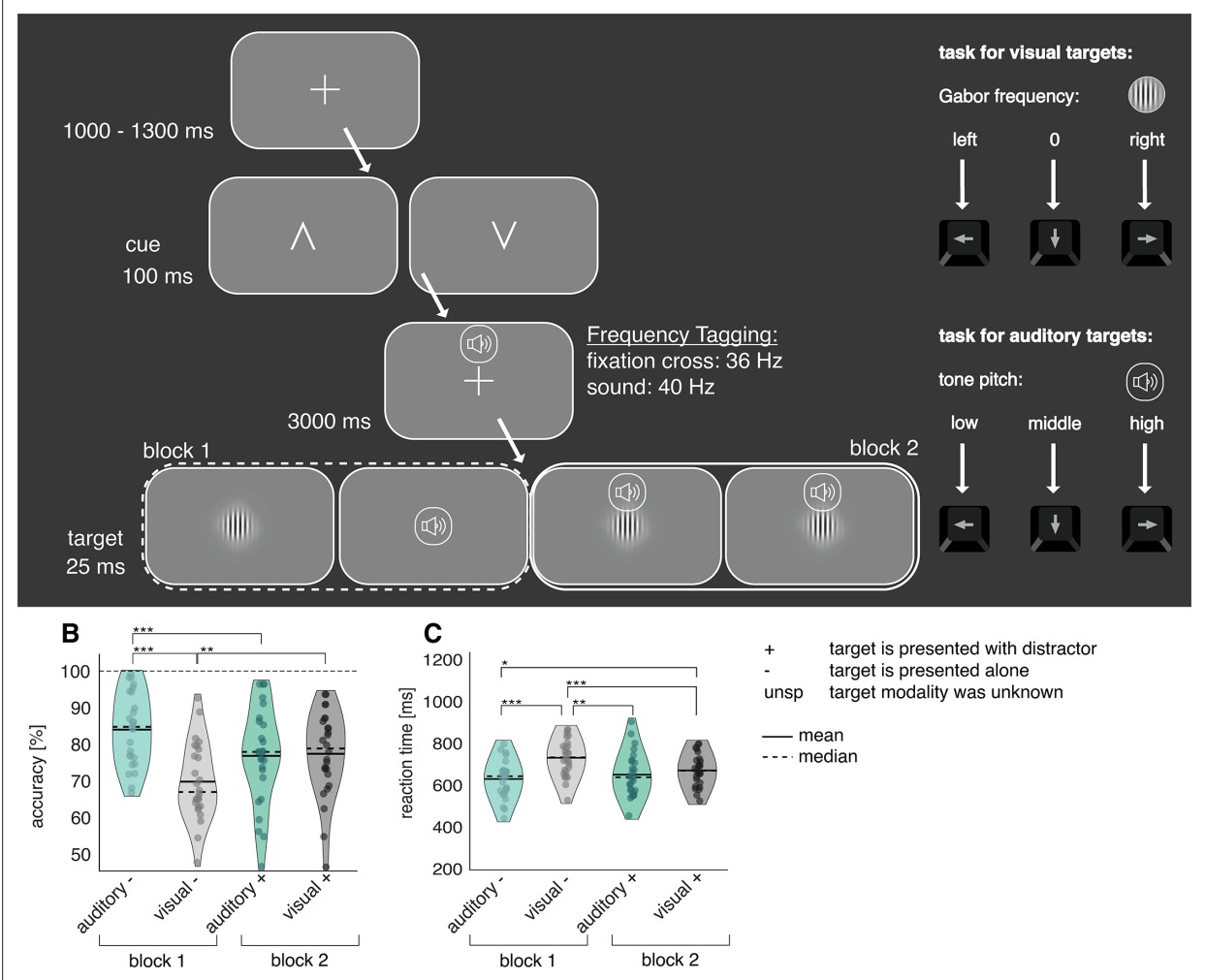

**Figure 7.** Illustration of the cross-model discrimination task in study 2. (**A**) Trials were separated by a 1–1.3 s interval, in which a fixation cross was displayed. A brief central presentation of the cue (100 ms) initiated the trial, signalising the target modality (from left to right: auditory, visual). In the cue-to-target interval, the fixation cross was frequency-tagged at 36 Hz. At the same time, a sound was displayed over headphones, which was frequency-tagged at 40 Hz. Both tones and fixation cross contained no task-relevant information. The target, consisting either of a static Gabor patch or a tone, was presented for 25 ms. Participants had to differentiate via button presses between three different Gabor stripe frequencies or tone pitches, respectively. In block 2, a distractor in the form of a random sound or Gabor patch of the un-cued modality was always presented alongside the target. Fixation was confirmed with an eye-tracker (see *Figure 7—figure supplement 1*). (**B, C**) Analysis of task accuracy and reaction time indicates comparable difficulties in block 2. (**B**) Task accuracy differences were only observable in the first block without distractors. (**C**) Reaction times to visual targets in the first block were strongly increased compared to all other conditions. In the second block, no significant difference in reaction times was observable. A distractor cost was found for accuracy as well as reaction times when expecting a visual target. When expecting an auditory target, distractor cost manifested mainly in reduced accuracy (see *Figure 2—figure supplement 1*). N=27; *** sig <0.001; ** sig.<0.01; * sig.<0.05.

The online version of this article includes the following figure supplement(s) for figure 7:

**Figure supplement 1.** Illustration of eye-tracking during the cue-to-target interval (2.5 – 0 s before target onset).

performed. Only in the easiest condition, when expecting a visual target that was not accompanied by a distractor, a negative correlation with reaction time could be found, strongest over right central electrodes (see *Figure 6B*; p=0.046). While a median split between slow and fast trials did not reach significance (p=0.50; $t_{(21)}$ = −0.69; $BF_{(10)}$ = 0.28), testing the correlation coefficients against 0 revealed strong evidence for a correlation (p=0.004; $t_{(21)}$ = −3.28; $BF_{(10)}$ = 11.98). If the intermodulation frequency reflects cross-modal integration (*Drijvers et al., 2021*), this effect would indicate faster reaction times for trials with stronger sensory integration.

## Study 2: Cross-modal MEG experiment

To test the robustness of our results and to employ additional control analyses, we replicated our experiment using MEG (see *Figure 7A*). While an increase in visual information processing parallel to an increase in alpha modulation already contradicts the notion of alpha inhibition exerting 'gain control', affecting the whole visual cortex, our claim that alpha modulation instead affects visual information at later processing stages still required further validation. As such, our goal was to perform source analyses showing alpha modulation originating from primary visual areas affected visual information at later processing stages (e.g. not in primary visual cortex). Additionally, to exclude that the uncertainty over possible distractors affected our results, we employed a block design, where block 1 consisted only of trials without distractors and in block 2 targets were always accompanied by a distractor. Furthermore, we aligned the visual and auditory task to be more similar, both of them now featuring frequency-discrimination, which related to sound pitch (frequency) in the auditory condition and stripe frequency of the Gabor patch in the visual condition. Lastly, to make sure our effects were driven by sensory modality differences rather than task-difficulty differences, we included a short calibration phase. Prior to the experiment, difficulty of pitch sounds and Gabor patch frequency were calibrated for each individual, ascertaining a success rate between 55% and 75%.

### Behavioural performance

Our adjustments in the MEG study streamlined performances to be more in line between auditory and visual conditions in the second block, however in the first block there were strong condition differences (Main effect condition: ($F_{(3,75)}$ = 10.26; p<0.01)). In block 1, participants were significantly more accurate in the auditory ('block 1 auditory', *M*=84% correct, SD = 10.03) compared to the visual task ('block 1 visual', *M*=70%, SD = 10.61; see *Figure 7B*). However, in block 2, there was no observable difference between visual and auditory task accuracy ('block 2 auditory', *M*=77% correct, SD = 10.03); ('block 2 visual', *M*=78%, SD = 13.15). Reaction times were comparable between conditions (overall: *M*=661 ms, SD = 76), with the exception of responses to visual targets in block 1 (block 1 visual: *M*=726 ms, SD = 85), which were significantly slower than all other reaction times (main effect condition: $F_{(3,75)}$ = 18.13; p<0.001; see *Figure 7C*).

Due to the block design, distractor cost could not be differentiated from learning effects. For auditory targets, there was a significant decrease in accuracy between block 1 and 2, while for visual targets, there was a significant increase. Similar to study 1, reaction times for auditory targets stayed roughly the same, but became shorter for visual targets (see *Figure 2—figure supplement 1*).

### Cross-modal cues differentially modulated pre-target alpha activity in early visual areas

In line with previous studies (*van Diepen and Mazaheri, 2017*), condition differences in alpha activity were only significant in block 2, where distractors were always present. As alpha modulation was a prerequisite to test our hypotheses, we performed the following analyses solely with data from block 2 (see *Figure 8*).

Replicating our results from study 1, cluster permutation on the Hilbert-transformed alpha activity revealed a significant cluster between −1.47 to −1.18 s (p=0.034), where alpha change from baseline was higher when expecting an auditory compared to a visual target (see *Figure 8A*). Applying roughly the cluster time-window (−1.5 and −1 s), we conducted a source localisation, contrasting the two conditions with cluster permutation analysis. We found that condition differences in alpha activity originated from early visual areas, albeit with a stronger effect on the right compared to the left hemisphere (p<0.01; peak spm coordinates: 41 −82−19 mm, in the right lingual gyrus see *Figure 8B*). Additionally, we replicated this effect with a virtual channel analysis in V1 (see *Figure 9—figure supplement 4*).

### Cross-modal cues increased the amplitude of the frequency-tagged responses across both modalities

In accordance with study 1, neuronal responses to 40 Hz auditory steady-state evoked field (ASSEF) responses were strongest over temporal areas (*Figure 9A*) and 36 Hz visual steady-state evoked field (VSSEF) responses were strongest over occipital areas (*Figure 9C*). As expected, the auditory tagging response originated from the right-hemispheric early auditory cortex (cluster significance:

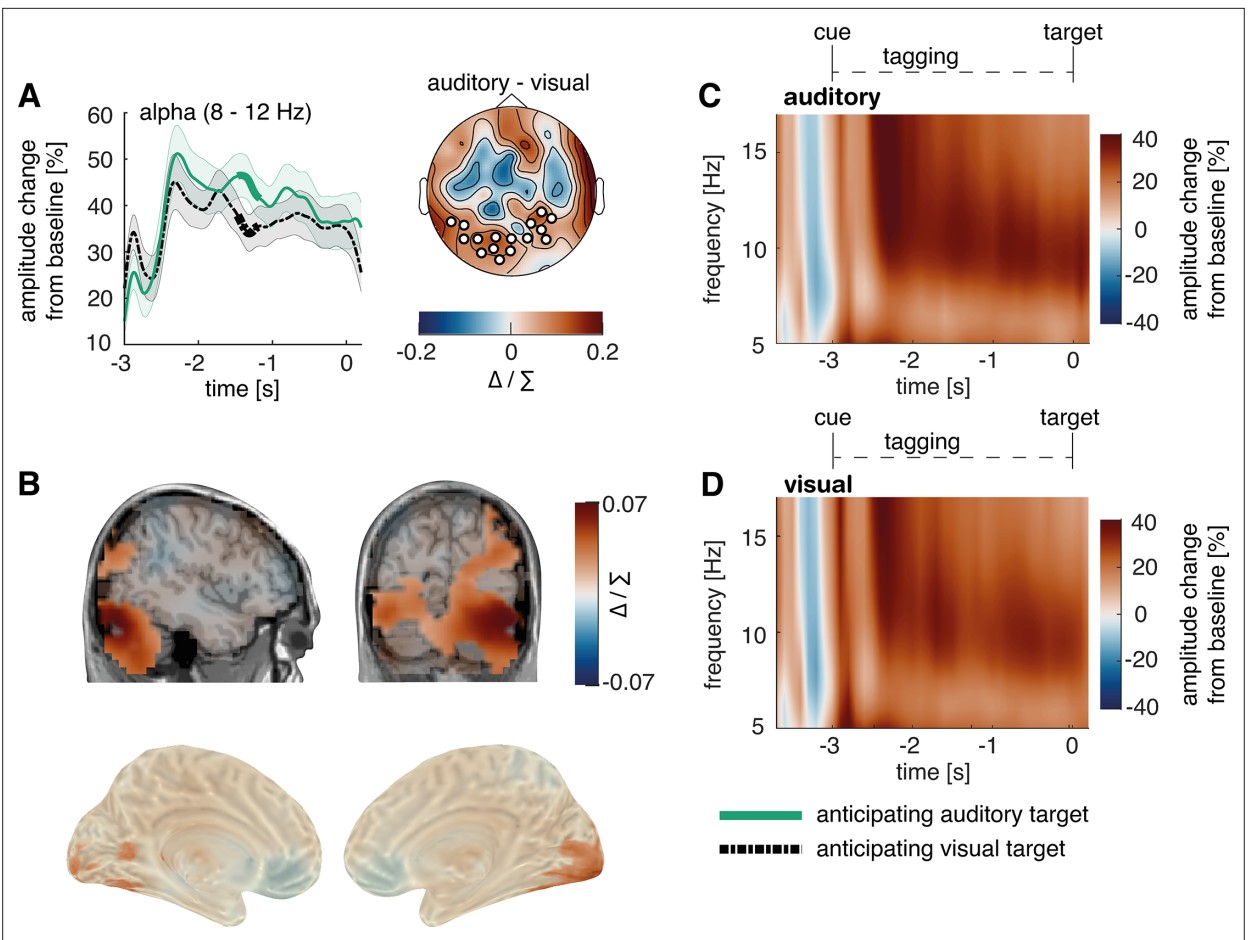

**Figure 8.** Post-cue modality-specific early visual modulation of alpha power in anticipation of an auditory versus a visual target. (**A**) The time course of post-cue alpha power. Cluster permutation analysis resulted in a condition effect, indicating heightened alpha activity when expecting an auditory compared to a visual target (p=0.034). We found the spectral power of alpha activity, however, not of 36 or 40 Hz to be different between conditions in this electrode cluster (see *Figure 8—figure supplement 1*). (**B**) Source localisation of the condition difference between expecting an auditory versus a visual target, revealing a significant cluster in early visual areas with stronger effects on the right hemisphere (p<0.01). (**C, D**) Time-frequency representation of power in the cue-to-target interval (average over electrodes that showed maximal condition difference in **A**). *Note:* Cluster electrodes are marked in white. Shading represents standard error from the mean; Δ / Σ represents (a-b)/(a+b) normalisation.

The online version of this article includes the following figure supplement(s) for figure 8:

**Figure supplement 1.** Power spectrum over MEG sensor clusters with significant condition differences.

p<0.01; peak spm coordinates: 69–22 5 mm, in the right superior temporal gyrus; see *Figure 9B*) and the visual tagging response originated from the early visual cortex (cluster significance: p<0.001; peak spm coordinates: 19 –105–11 mm, in the right lingual gyrus; see *Figure 9D*). Additionally, we observed significantly reduced 40 Hz activity in the left-hemispheric visual-to-central cortex (cluster significance: p<0.01).

Applying cluster permutation on the Hilbert-transformed frequency-tagging data over the last 500 ms before target onset replicated the results of study 1, where both 36 Hz VSSEF as well as 40 Hz ASSEF signals were significantly enhanced when expecting an auditory compared to a visual target (see *Figure 9E–G*; auditory target: p=0.043; visual target: p=0.019).

Source localisation confirmed the condition difference in 36 Hz VSSEF activity to originate from later stages of the processing stream, encompassing a wide range of areas, most strongly the medial occipital cortex and the precuneus (cluster significance: p=0.047; peak spm coordinates: 3–66 44 mm, in the left and right precuneus; see *Figure 9H*). In source space, the effect was not significant for auditory 40 Hz activity (p=0.11; peak spm coordinates: –9–39 0 mm, in the left precuneus; see *Figure 9F*).

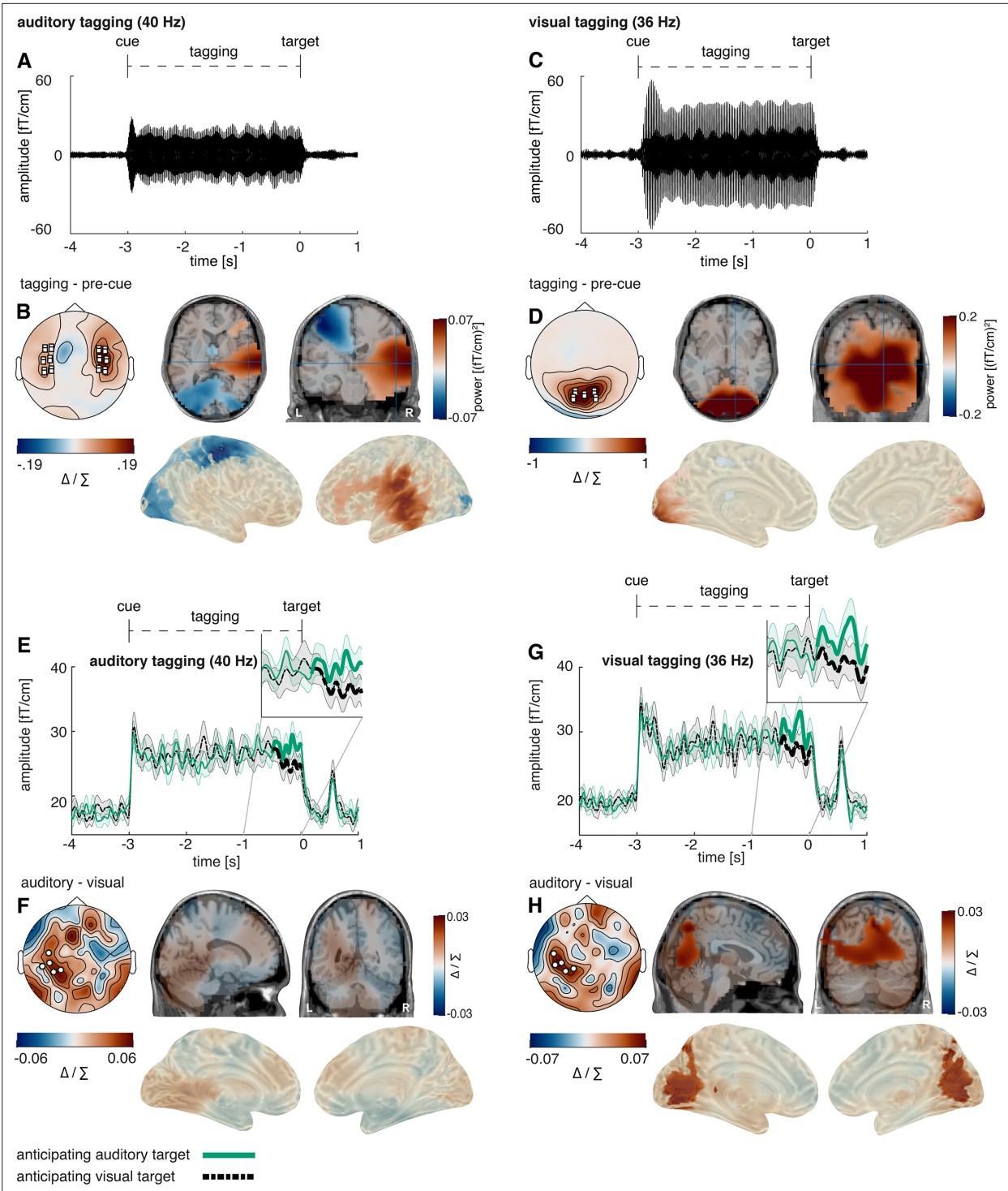

**Figure 9.** Post-cue modality-specific early visual modulation of alpha power in anticipation of an auditory versus a visual target. (**A**) Auditory steady-state evoked fields (ASSEF) averaged over 24 temporal sensors displaying the highest auditory 40 Hz power (12 right, 12 left; marked in white squares). (**B**) ASSEF source localisation revealed a significant positive cluster in the right-hemispheric early auditory cortex (p<0.001). (**C**) Visual steady-state evoked fields (VSSEF) averaged over 10 occipital sensors, displaying the highest visual 36 Hz power (marked in white squares). (**D**) VSSEF source localisation revealed a significant positive cluster in the early visual cortex (p<0.001). (**E**) The Hilbert envelope of the 40 Hz ASSEF reveals an increase shortly before target onset when anticipating an auditory compared to a visual target (p=0.043); we found the spectral power of 36 Hz and 40 Hz power, but not of alpha power, to be different between conditions in this cluster (*Figure 8—figure supplement 1*). To confirm that 4 Hz difference between the tagged frequencies was sufficient to prevent bleeding over of effects between conditions, we also analysed the Hilbert envelope for 44 Hz when expecting

*Figure 9 continued on next page*

Figure 9 continued

an auditory target and found no significant condition difference (see *Figure 9—figure supplement 1*). (**F**) Condition differences in the 40 Hz ASSEF response did not reach significance in sensor space. (**G**) The Hilbert envelope of the 36 Hz VSSEF likewise reveals an increase shortly before target onset when anticipating an auditory compared to a visual target (p=0.019). (**E–G**) For individual power spectra across conditions, see *Figure 9—figure supplements 2 and 3*. In an additional virtual channel analysis, we found no significant difference between expecting a visual and an auditory target for 36 Hz activity in V1 and for 40 Hz activity in Heschl's Gyrus (see *Figure 9—figure supplement 4*). (**H**) Condition differences in the 36 Hz VSSEF response were significant over several areas of the visual stream, including most strongly the medial occipital cortex, the calcarine fissure and the precuneus (p=0.047); *Note*: Cluster electrodes are marked in white. Shading represents standard error from the mean. Δ / ∑ represents (a-b)/(a+b) normalisation.

The online version of this article includes the following figure supplement(s) for figure 9:

**Figure supplement 1.** Illustration of bleeding over effects over a span of 4 Hz.

**Figure supplement 2.** Individual ERP power spectra of the cue-to-target interval when anticipating an auditory target in the MEG-study.

**Figure supplement 3.** Individual ERP power spectra of the cue-to-target interval when anticipating a visual target in the MEG-study.

**Figure supplement 4.** Virtual channels for V1 and Helschl's gyrus.

---

Furthermore, a virtual channel analysis in V1 and Heschl's gyrus confirmed that there were no condition differences in primary visual and auditory areas (see *Figure 9—figure supplement 4*).

## Alpha power was positively correlated with amplitude of frequency-tagged responses

The positive correlations between alpha power and frequency-tagging amplitude observed in study 1 could be replicated. Alpha activity 500 ms before target onset correlated both with 36 Hz VSSEF and 40 Hz ASSEF activity when expecting a visual target (see *Figure 10A and B*, 36 Hz response: cluster significance: p<0.01; median split: p<0.001; $t_{(24)} = 4.33$; $BF_{(10)} = 127$; t-test: p<0.001; $t_{(24)} = 5.33$; $BF_{(10)} = 1272$; 40 Hz response: cluster significance: p<0.001; median split: p<0.001; $t_{(24)} = 7.05$; $BF_{(10)} = 59443$; t-test: p<0.001; $t_{(24)} = 6.75$; $BF_{(10)} = 30515$).

The same correlation was present when expecting an auditory target for both 36 Hz VSSEF activity (see *Figure 10D and E*; cluster significance: p<0.001; median split: p=0.001; $t_{(23)} = 3.62$; $BF_{(10)} = 25.51$; t-test: p<0.001; $t_{(23)} = 4.60$; $BF_{(10)} = 216$) and 40 Hz ASSEF activity (cluster significance: p=0.005; median split: p<0.001; $t_{(25)} = 3.75$; $BF_{(10)} = 36.40$; t-test: p<0.001; $t_{(25)} = 4.06$; $BF_{(10)} = 73.61$).

In accordance with the results of study 1, we then tested whether alpha activity preceding frequency-tagging activity showed similar correlations. Our data revealed that during the last 1–1.5 s before target onset alpha activity correlated with the 36 Hz VSSEF response during the last 500 ms prior to an auditory target (see *Figure 10F*; cluster significance: p=0.01; median split: p<0.001; $t_{(24)} = 4.66$; $BF_{(10)} = 271$; t-test: p<0.001; $t_{(24)} = 5.66$; $BF_{(10)} = 2688$). The same alpha activity correlated with 40 Hz ASSEF activity during the last 500 ms prior to a visual target (see *Figure 10C*; cluster significance: p=0.002; median split: p=0.002; $t_{(23)} = 3.41$; $BF_{(10)} = 16.38$; t-test: p<0.001; $t_{(23)} = 5.57$; $BF_{(10)} = 1901$).

Lastly, both alpha activity as well as 36 Hz VSSEF responses 500 ms before target onset correlated negatively with reaction time on a trial-by-trial basis, indicating faster reaction times in trials with higher pre-stimulus activity (alpha: p=0.037; median split: p=0.013; $t_{(25)} = -2.67$; $BF_{(10)} = 3.78$; t-test: p<0.01; $t_{(25)} = -3.34$; $BF_{(10)} = 14.84$; 36 Hz: p=0.002; median split: p=0.004; $t_{(25)} = -3.20$; $BF_{(10)} = 8.98$; t-test: p<0.01; $t_{(25)} = -3.46$; $BF_{(10)} = 19.12$. See *Figure 10—figure supplements 2 and 3*).

The same increase in the intermodulatory frequency observed in study 1 could be observed in our second study, during the last 500 ms prior to target onset (see *Figure 11A*; p=0.006). In source space, a descriptive condition difference was visible in auditory and visual sensory cortices; however, this effect did not reach significance (p=0.49; peak spm coordinates: 45 –83–19 mm, in the right lingual gyrus; see *Figure 11B*).

## Discussion

The neuropsychological account of attention defines it as the selective facilitation (i.e. prioritisation) of relevant sensory input and suppression of irrelevant sensory input. Oscillatory activity in the alpha range (~10 Hz) has been suggested to play a mechanistic role in attention through inhibition of irrelevant cortices, commonly referred to as the 'alpha inhibition hypothesis' (*Foxe et al., 1998*; *Jensen and Mazaheri, 2010*; *Klimesch et al., 2007*). In the current cross-modal attention study, we directly

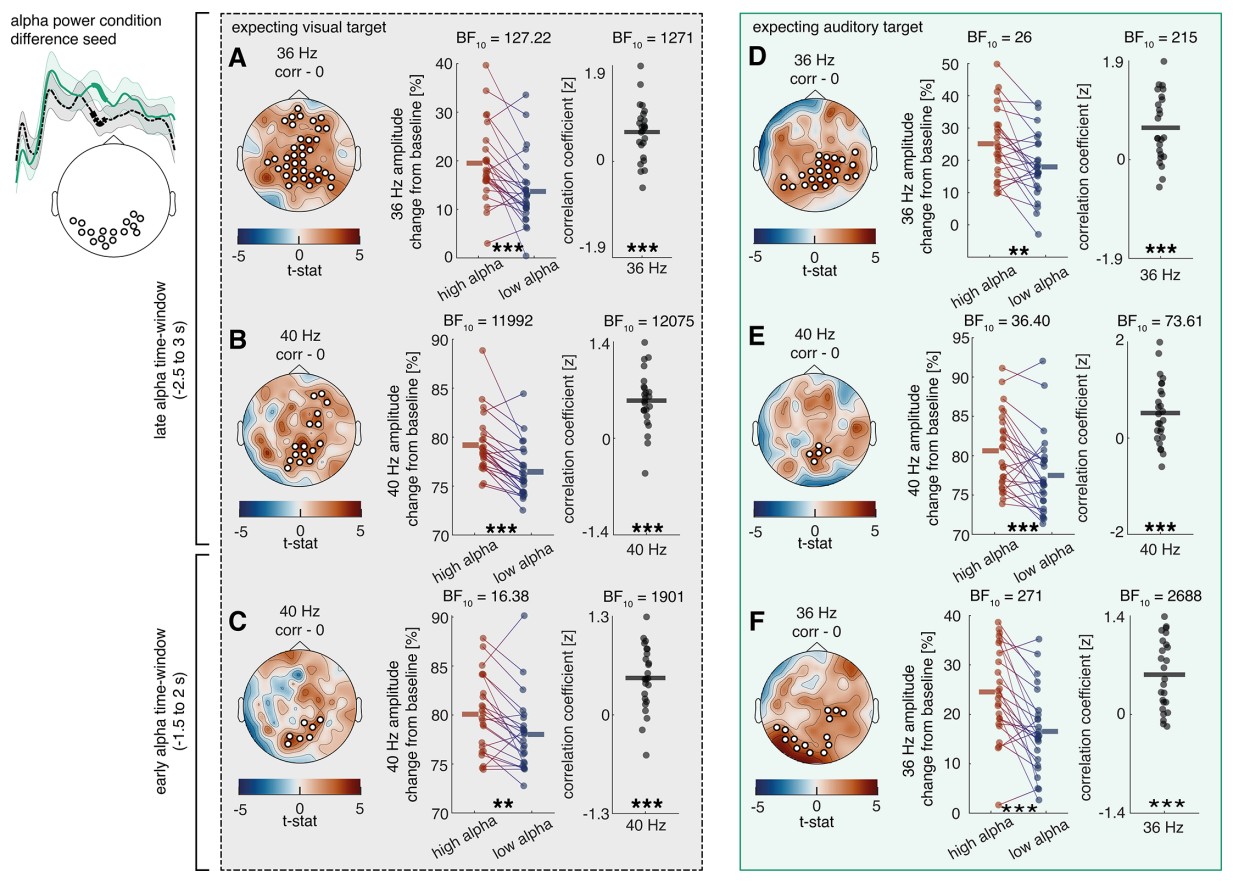

**Figure 10.** Relationship between cue-induced alpha modulation and amplitude of frequency-tagged responses in study 2. (**A–C**) A positive correlation is visible between alpha activity in the last 500 ms as well as alpha activity in the last 1500–1000 ms and steady state potentials shortly before target onset when expecting a visual target (36 HZ late: p=0.013; 40 Hz late: p=0.009; 40 Hz early: p=002 popp). For an exemplary illustration of individual correlations between alpha activity and 36 Hz activity shown in (**A**), see *Figure 10—figure supplement 1*. (**D–F**) when expecting an auditory target, there is a positive correlation between alpha activity in the last 500 ms as well as alpha activity in the last 1500–1000 ms and steady state potentials shortly before target onset (36 HZ late: p<0.001; 40 Hz late: p=0.005; 36 Hz early: p=011 popp). Furthermore, we found a significant correlation between alpha activity as well as 36 Hz SSVEFs with reaction time (see *Figure 10—figure supplements 2 and 3*). Additionally, we showed that for the last second before target onset, we found significant correlations between the alpha cluster (localised to early visual areas) and 36 Hz activity (over a separate cluster) when expecting an auditory target. When correlating alpha activity with 36 Hz activity within the same early visual electrode cluster, no clear effect was found (see *Figure 10—figure supplement 4*) N=27; *** sig <0.001; ** sig.<0.01; * sig.<0.05.+sig.<0.1.

The online version of this article includes the following figure supplement(s) for figure 10:

**Figure supplement 1.** Exemplary illustration of the correlation between alpha power (0.5 – 0 s before target onset) and 36 Hz steady-state response (0.5 – 0 s before target onset) for each participant in block 2 (distractors present) when expecting a visual target.

**Figure supplement 2.** Correlation of prestimulus alpha change from baseline with reaction time in the MEG study.

**Figure supplement 3.** Correlation of 36 Hz change from baseline with reaction time in the MEG study.

**Figure supplement 4.** Relationship between cue induced alpha modulation and amplitude of frequency tagged responses in the MEG study, both inter- and intra-cortically.

tested this hypothesis by using frequency-tagging to specifically examine how cues signalling the modality of an upcoming target (either the auditory or visual modality) affected the responsiveness of the relevant and irrelevant sensory cortices prior to target onset. In-line with previous work, we observed a post-cue increase in posterior alpha power in anticipation of processing auditory targets. However, contrary to prevalent theories proposing visual gain suppression when focusing on the auditory modality, we observed that the amplitude of visual frequency-tagging responses increased just prior to the onset of the auditory target. This suggests that responsiveness of the visual stream was not inhibited when attention was directed to auditory processing and was not inhibited by occipital alpha activity, which directly contradicts the proposed mechanism behind the alpha inhibition

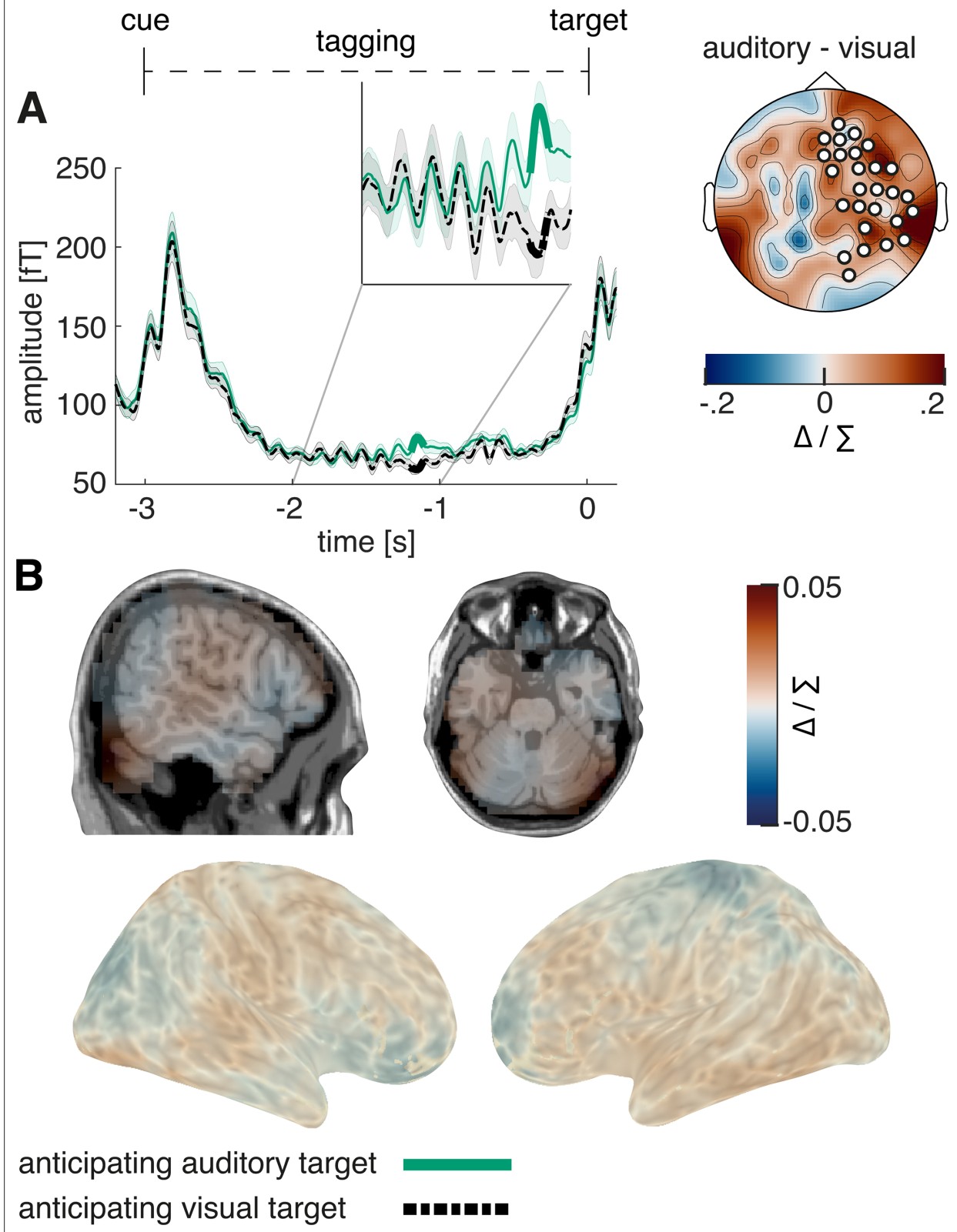

**Figure 11.** Steady-state response in the intermodulation frequency. (**A**) Similar to study 1, 4 Hz frequency-tagging activity was increased shortly before target onset when expecting an auditory compared to a visual target (p=0.006). (**B**) Source localisation showed activity over auditory sensory areas, but did not reach significance. N=27.

hypothesis. Our results reconcile previously paradoxical results on audio-visual attention and support the view that alpha activity gates downstream communication pathways.

## Frequency-tagging

In the current experiment, we specifically chose to analyse the cue-to-target interval, where both visual and auditory SSEPs/SSEFs present preparatory states for the upcoming task, independent of task-related processing or performance. The magnitude of auditory SSEPs/SSEFs was increased shortly before target onset when expecting an auditory target compared to a visual target, very much in line with previous reports (e.g., *Saupe et al., 2009*). In contrast to the results reported in *Saupe et al., 2009*, where visual SSEPs decreased when attending the auditory modality, visual SSEPs/SSEFs increased shortly before target onset when expecting an *auditory* target in our data. This is especially surprising as auditory targets were frequently, or in the case of our second study, always accompanied by visual distractors, rendering it optimal for task success to completely ignore any visual input. As auditory targets were significantly more difficult than visual targets in our first study and of comparable difficulty in our second study, these results strongly speak to a vigilance increase of sensory processing independent of modality and an inability to selectively disengage one sensory modality in anticipation of a demanding task. This view is consistent with previous work in which visual SSEPs elicited by irrelevant background stimulation increased with task load in an auditory discrimination task (*Jacoby et al., 2012*). Furthermore, our results indicate that task demand is a strong candidate to reconcile previously seemingly paradox results, as splitting attention between the auditory and visual system seemed possible in simpler tasks (*Driver and Spence, 1998*; *Saupe et al., 2009*) and impossible under high demand (*de Jong et al., 2010*; *Driver, 1996*; *Driver and Spence, 1998*; *Spence and Driver, 1996*). An alternative account for our findings stems from the evidence that participants are more likely to only perceive and react to the visual modality when confronted with audio-visual stimuli (*Colavita, 1974*; *Spence, 2009*). However, this effect was mostly limited to speeded modality discrimination/target detection tasks (*Sinnett et al., 2008*; *Spence, 2009*). Furthermore, the increased difficulty of the here-used auditory stimuli was confirmed in a previous block-design study (*van Diepen and Mazaheri, 2017*) and in our second study, performances over the visual and auditory tasks were comparable. Nevertheless, visual dominance could play a role for auditory target difficulty as well as predictions over the reciprocity of the audio-visual relationship.

## A revision of the alpha inhibition hypothesis

Top-down cued changes in alpha power have now been widely viewed to play a functional role in directing attention: the processing of irrelevant information is attenuated by increasing alpha power in areas involved with processing this information (*Foxe et al., 1998*; *Klimesch et al., 2007*; *Jensen and Mazaheri, 2010*). However, recent evidence suggests that posterior alpha activity does not inhibit gain in early sensory processing stages (*Antonov et al., 2020*; *Gundlach et al., 2020*; *Gutteling et al., 2022*; *Zhigalov and Jensen, 2020*). To date there has been no direct investigation into the effect of alpha increases on later stages in the processing stream. In the current study, as expected, we observed a post-cue increase in occipital alpha activity in anticipation of an auditory target. However, we also observed an increase in the amplitude of visual SSEPs during the cue-target interval, which directly contradicts the widespread view of alpha activity exerting 'gain control' in early sensory areas by regulating excitability (*Foxe and Snyder, 2011*; *Jensen and Mazaheri, 2010*; *Van Diepen et al., 2019*). Here we propose that alpha activity, rather than modulating early primary sensory processing, exhibits its inhibitory effects at later stages of the processing stream (*Antonov et al., 2020*; *Gundlach et al., 2020*; *Zhigalov and Jensen, 2020*; *Zumer et al., 2014*), gating feedforward or feedback communication between sensory areas (*Bauer et al., 2020*; *Haegens et al., 2015*; *Uemura et al., 2021*). Our data provides evidence in favour of this view, as we can show that early sensory alpha activity does not covary over trials with SSEP magnitude in early visual areas, but covaries instead over trials with SSEP magnitude in higher order sensory areas (see also *Figure 10—figure supplement 4*). If alpha activity exerted gain control in early visual regions, increased alpha activity would have to lead to a decrease in SSEP responses. In contrast, we observe that increased alpha activity originating from early visual cortex is related to *enhanced* visual processing. Source localisation confirmed that this enhancement was not originating from early visual areas, but from areas associated with later stages of the processing stream such as the precuneus, which has been connected to sensory

integration (*Al-Ramadhani et al., 2021*; *Xie et al., 2019*). While we cannot completely rule out alternative explanations, it seems plausible to assume that inhibition of other task-irrelevant communication pathways leads to prioritised and thereby enhanced processing over relevant pathways. In line with previous literature (*Morrow et al., 2023*; *Peylo et al., 2021*; *Zhigalov and Jensen, 2020*), we therefore suggest that alpha activity limits task-irrelevant feedforward communication, thereby enhancing processing capabilities in relevant downstream areas (see *Figure 1A*). The benefit of alpha-modulation and its effect on visual information processing is underlined by reaction times, which were faster both for trials with high pre-target alpha activity and high pre-target visual SSVEF activity.

It should be noted that the comparison between modulation in alpha activity and in SSEP/SSEFs is difficult, especially concerning timing. This is largely owed to differences in signal-to-noise due to trial averaging in the frequency versus the time domain and temporal and frequency lag in the estimation of alpha activity (*Peylo et al., 2021*). It is further noteworthy that the majority of evidence for the alpha inhibition hypothesis focused on the effect of pre-target alpha modulation on behaviour and target-related potentials (*Morrow et al., 2023*). However, in our data alpha modulation occurs clearly ahead of SSVEP/SSVEF modulation on a scale that could not be simply explained by temporal or frequency smearing. Additionally, significant trial-by-trial correlations, which occur in the frequency domain for both signal types, underline the strong relationship between both measurements.

Interestingly, we could show that the magnitude of the correlation between alpha power and visual information processing varied between conditions, suggesting a dynamic and adaptive regime. This notion supports the view that alpha oscillations represent a mechanism rather than a specific function, which can fulfil different roles depending on task demand and network location, which has been confirmed in a recent study revealing functionally distinct alpha networks (*Clausner et al., 2024*). As such, it is conceivable that alpha oscillations can in some cases inhibit local transmission, while in other cases, depending on network location, connectivity and demand, alpha oscillation can facilitate signal transmission. This mechanism allows to increase transmission of relevant information and to block transmission of distractors. In different contexts, utilising unimodal targets and distractors, spatial cueing, or covert attention, different functional processes could be involved (*Morrow et al., 2023*). Future research should intensify efforts to disentangle these effects, investigating localised alpha networks intracranially or through combinations of fMRI, EEG and MEG, to clearly measure their effects on sensory processing and behaviour.

It is known that the localisation of alpha activity reflects the retinotopic organisation of visual spatial attention topographically over the parietooccipital cortex (*Kelly et al., 2006*; *Popov et al., 2019*). Notably, recent studies provided evidence that the same organisation can be observed for auditory attention. Specifically, the localisation of visual alpha activity in the parietooccipital cortex reflects the spatial direction of auditory attention (*Klatt et al., 2021*; *Popov et al., 2021*). This observation can be explained through micro-saccades towards the spatial location of sounds, which are irrevocably connected to alpha oscillations (*Popov et al., 2021*). While we did not manipulate spatial attention, our results fit well to the notion of visual alpha activity serving as a sensory orientation system, relaying visual information to task-relevant downstream processing areas, and blocking communication to irrelevant pathways.

## The intermodulation frequency

Previous research showed that simultaneous frequency-tagging in multiple frequencies evokes a response in the intermodulation frequency (f1–f2). In multimodal settings, this frequency is thought to reflect cross-modal integration (*Drijvers et al., 2021*). This is very well in line with our findings, where increased vigilance of the sensory system arising from anticipation of a difficult auditory target resulted in an increase in the intermodulation frequency. Likewise, our data shows that visual signal enhancement was localised in precuneus, which has been connected to sensory integration (*Al-Ramadhani et al., 2021*; *Xie et al., 2019*). Furthermore, we could show that the intermodulation frequency covaries over trials with reaction time in the easiest condition, where visual targets were presented without any distractors. A lack of this connection in other conditions might reflect increasing interferences from higher task difficulty, rather than a lack of the effect itself, but this remains to be tested. We cannot exclude an alternative explanation, as theta oscillations are known to be involved in movement preparation, it is possible that phase-resets could lead to time-locked appearance of these oscillations (*Lakatos et al., 2008*; *Tomassini et al., 2017*).

## Conclusion

Our results taken together suggest that under high attention, audio-visual excitability is enhanced, reflecting an increase in vigilance for both visual as well as auditory information, even if this increases processing of distracting information. We showed that this vigilance shift, as reflected by SSEP/SSEF responses, is regulated by early visual alpha activity, presumably through relaying of visual information over communication pathways, thereby controlling the downstream flow of visual information.

# Materials and methods

### Key resources table

| Reagent type (species) or resource | Designation | Source or reference | Identifiers | Additional information |
|---|---|---|---|---|
| Software, algorithm | MATLAB | Mathworks | R2020b | https://de.mathworks.com/products/matlab.html |
| Software, algorithm | Psychtoolbox 3 | Psychtoolbox 3 | 3.0.19 | http://psychtoolbox.org |
| Software, algorithm | Fieldtrip | Fieldtrip | 20251106 | https://www.fieldtriptoolbox.org |

### Participants EEG-study

In total, 24 healthy volunteers participated in this study (mean age: 19.1±1.8 SD; 17 women). Due to technical difficulties, one participant could not finish the experiment and one participant did not exceed chance level in the behavioural task (~33%). Both were therefore removed from any further analysis. All remaining participants reported normal or corrected-to-normal vision, no history of psychiatric or neurological illness and provided written informed consent. After completion of the experiment, participants received either monetary compensation or certification of their participation for their university course programme. The study protocol was approved by the Ethics Committee of the School of Psychology at the University of Birmingham and is in accordance with the Declaration of Helsinki.

### Participants MEG-study

In total, 28 healthy volunteers participated in this study (mean age: 23.4±3.6 SD; 20 women). One participant was removed from further analysis, as they only responded to ~42% of trials correctly in the second block, which related to 27/19 trials per condition, respectively. All remaining participants reported normal or corrected-to-normal vision, no history of psychiatric or neurological illness and provided written informed consent. After completion of the experiment, participants received either monetary compensation or certification of their participation for their university course programme. The study protocol was approved by the Ethics Committee of the School of Psychology at the University of Birmingham and is in accordance with the Declaration of Helsinki.

### Cross-modal attention paradigm EEG-study

Each trial was initiated by a brief presentation of a cue (100 ms) signalling the modality of the upcoming discrimination task (v-shape: visual modality; inversed v-shape: auditory modality; diamond-shape: non-specific). During the following cue-to-target interval, the fixation cross was frequency-tagged at 36 Hz. At the same time, a 40 Hz frequency-tagged sound (amplitude modulated white noise) was played over headphones (see *Figure 1*). The volume of tones was initially adjusted to a level that was clearly perceivable but not uncomfortable and remained stable over participants. No task was connected to this interval and participants did not need to pay attention to either the sound or the fixation cross. After three seconds, the frequency-tagging stopped, and right after the cessation of stimuli, the target was presented for a very brief moment (25 ms). It consisted either of a Gabor patch (visual modality) or a sound (auditory modality). If the target modality was visual, participants had to use the three arrow buttons on the keyboard to indicate whether the Gabor patch was tilted to the left (–10°; left arrow button), vertical (0°; down arrow button), or tilted to the right (10°; right arrow button). Additionally, in 50% of the trials, a random distractor from the pool of auditory targets was presented simultaneously to the visual target over headphones. Similarly, if the target modality was auditory, participants used the same buttons to indicate whether the pitch of the tone was low (500 Hz), medium (1000 Hz), or high (2000 Hz). Again, in 50% of the trials, a random distractor from the pool of visual targets was also presented. If the cue was non-specific (diamond shape), either a

visual (50% of non-specific trials) or an auditory target was presented, never accompanied by any distractors. Experimental trials were separated by an inter-trial interval of 4 seconds, to avoid carry-over effects from previous trials. The resulting 6 conditions were randomly ordered and balanced out over the experiment.

During the experiment, participants were instructed to keep their gaze locked to a fixation cross presented at the centre of the screen. Preceding data collection, participants performed 36 practice trials to get accustomed to the task and the target stimuli. The ensuing experiment was split into 26 trial sequences, separated by self-chosen breaks, which together resulted in 468 trials and lasted between 80 and 90 minutes. The discrimination task was programmed and presented with MATLAB R2020b and Psychtoolbox-3 on an LCD-monitor featuring a 140 Hz refresh rate. The onset of the visual and auditory tagging frequencies (i.e steady state stimuli) were tracked using the Cedrus Stimtracker (https://cedrus.com/stimtracker/index.htm).

## Adjustments to the attention paradigm in the MEG-study

In our second study, we removed all ambiguity concerning targets and distractors and therefore developed a blocked design, incorporating two blocks. The first block did not display distractors and only cues which correctly predicted the target were presented. Cues in the second block were likewise always correctly indicating the target modality, but this time, each target was accompanied by a random distractor from the non-target modality.

Furthermore, the visual task was adjusted to be more in line with the auditory task. As such, the Gabor patches now featured stripes in different frequencies (e.g. a low number of stripes, a medium number of stripes and a high number of stripes). The participant's task was to discriminate between these three Gabor patches. As auditory targets had been markedly more difficult in our first study, we now included a brief difficulty calibration prior to the experiment. First, we presented 21 Gabor patches with three different amounts of stripes following a standard difficulty. If participants could discriminate them correctly 55–75% of the time, this difficulty setting was chosen. Otherwise, depending on the performance, the stripe-frequency of the Gabor patches was adjusted. There were maximally three sessions of 21 Gabor patches, after which we had enough data to calibrate the individual difficulty setting.

The same procedure was then performed with the difficulty of the tones, calibrating the pitch frequency for each individual participant.

Lastly, visual frequency-tagging stimulation now followed a sinusoidal contrast-change rather than an on-off stimulation, which was possible due to a high-resolution projector featuring a refresh rate of 1440 Hz (PROPixx DLP LED projector;VPixx Technologies Inc, Canada).

## Eye-tracking

To confirm that participants had focused on the fixation cross during the cue-to-target interval, we incorporated eye-tracking into our MEG-experiment (EyeLink 1000 Plus). Correct trials of the second block were analysed for vertical and horizontal eye-movements. To exclude blinks from this analysis, trials with very large eye-movements (>10 degrees of visual angle) were removed from the eye-tracking data (See *Figure 7—figure supplement 1*).

## Behavioural analysis

We were interested in accuracy in the discrimination of visual targets and auditory targets, as well as reaction times. Furthermore, we examined the distraction cost of having a target presented with a distractor of a different modality as well as the reaction time to make the target discrimination. The distraction cost was calculated as the reaction time difference between cued targets with distractors (i.e. visual and auditory stimuli presented together) and cued targets without distractors (either a visual or auditory stimulus presented alone). All incorrect trials as well as trials with reaction times faster than 100 ms or exceeding 1500 ms were removed from analysis (0.5% too fast, 9.2% too slow).

## EEG data acquisition

All EEG recordings were conducted using a WaveGuard Cap (ANTneuro), featuring 64 Ag/AgCl electrodes (10–10 system; ground: Fz; reference: Cpz; EOG: left canthus). Electrodes positions were prepared with OneStep cleargel conductive paste and impedances were kept below 100 kΩ. The

measured signal was transmitted using an ANTneuro EEGosports amplifier (low-pass filter: 150 Hz; high-pass filter: 0.5 Hz; sampling rate: 500 Hz).

## MEG data acquisition

Prior to the experiment, feducial positions and head-shape were recorded using a FASTRAK system (Polhemus, USA). The experiment took place in a dimly lit room, where participants were seated in a comfortable chair in the gantry of a 306-sensor TRIUX Elekta system with 204 orthogonal planar gradiometers and 102 magnetometers (Elekta, Finland). The 71 * 40 cm screen was positioned at ~1.40 m distance from the participant.

## EEG preprocessing

Offline analyses were performed in MATLAB R2020b. The data was pruned from artifacts by visual inspection using the EEGLAB toolbox (*Delorme and Makeig, 2004*). Additionally, blinks and ocular artefacts were removed from the data using independent component analysis (ICA). EEG channels were re-referenced to an average of all channels (excluding EOG).

## MEG preprocessing

Offline analyses were performed in MATLAB R2020b and Python. Spatiotemporal Signal-Source-Separation (SSS) was applied to the raw data via MNE's inbuilt maxfilter function with a duration window of 10 s and a correlation value of 0.9. The data was pruned from artifacts by visual inspection using the Fieldtrip toolbox (*Oostenveld et al., 2011*). Additionally, blinks and ocular artefacts were removed from the data using ICA. In sensor space, planar gradiometers were combined for further analyses. In source space, all individual planar gradiometers were analysed.

## Amplitude of the evoked frequency-tagging response

To investigate the temporal dynamics of amplitude of the frequency-tagged responses after the onset of the attentional cues (also the precise onset of the frequency-tagged stimuli) the data was epoched into 6-second segments starting 1.5 seconds prior to cue onset. Next the data were narrow-band filtered around the 36 Hz activity to capture the visual frequency-tagging, 40 Hz activity in to capture the auditory frequency-tagging, and the intermodulation frequency at 4 Hz, which can be derived by subtracting both frequency-tagging responses ($f_i = f_{auditory} - f_{visual}$; see *Drijvers et al., 2021*). Here we used a Blackmann-windowed sync filters adapted to a suitable ratio of temporal and frequency resolution for the specific frequency of each of the tagged signals: filter order 116 for 35.5–36.5 Hz and 39.5–40.5, filter order 344 for 3.5–4.5 Hz. The filtered data at each of the tagged frequencies as well the intermodulated frequency were baseline corrected (interval between 700 and 200 ms preceding cue onset) before calculating the average over trials to obtain steady-state evoked potentials. The power envelope of the SSEPs of tagged frequencies was estimated using Hilbert transformation. To confirm that 4 Hz is a sufficient distance between tagging frequencies, we repeated the analysis for 43.5–44.5. We found no indication of frequency-bleeding over, as the effects observed at 40 Hz were not present at 44 Hz (see *Figure 9—figure supplement 1*).

## Temporal dynamics of the induced EEG changes

In addition to looking at the cue evoked changes in the amplitude of the frequency-tagged signals, we investigated the induced changes in the EEG signal at the frequencies of the tagged auditory and visual stimuli, alpha activity (9–11 Hz, filter order 276 for 7.5–12.5 Hz), as well as the intermodulation frequency (4 Hz; filter order 344 for 3.5–4.5 Hz). Here rather than averaging the epoched data filtered at the specific frequency ranges, we performed the Hilbert transform, and averaged the power-envelope of the specific frequencies across trials. This approach is very much analogous to the standard time-frequency analysis using convolutions (*van Diepen and Mazaheri, 2017*; *Zhigalov et al., 2019*), but affords more control concerning temporal versus frequency resolution to examine the temporal dynamics of the specific frequencies of interest. In our second study, the individual peak alpha frequency was used for bandpass-filtering in contrast to a standardised band applied in the first study.

## Time-frequency representations of power

In addition to estimating frequency power envelopes, time–frequency representations (TFRs) of power of the EEG signal were estimated using the Fieldtrip toolbox (*Oostenveld et al., 2011*). The power or frequencies between 5 and 20 Hz were calculated for each trial, using a sliding time window (frequency steps: 0.5; time steps: 10 ms). The length of the window was adjusted to a length of 3 cycles per frequency and tapered with a Hanning window. For each trial, both datasets were normalised to display relative percent change from baseline using the following formula: [(activity – baseline) / baseline], where baseline refers to the interval between 700 and 200 ms before cue onset. To estimate the topographical distribution of voltage differences between conditions, uncorrected power values were normalised applying the following formula: $[\Delta/\sum = (a - b) / (a+b)]$, where a and b reflect the different conditions.

## Source localisation

Source localisation was performed with a beamformer approach using the Fieldtrip toolbox. Head-models were created based on individual T1-scans fitted to fiducial points and head shapes. These data were fit to a 5 mm 3D source model and warped into MNI-space. Two participants were missing individual T1-scans. In these cases, we applied a standardised T1-scan using the Colin 27 Average Brain Model (*Holmes et al., 1998*). Frequency-domain data was localised using the Dynamic Imaging of Coherent Sources (DICS beamformer) method with a dpss taper of 2 Hz assuming fixed orientation. As condition differences between frequency-tagging responses were better estimated in the time-domain, we assessed them applying the Synthetic Aperture Magnetometry (SAM-beamformer) method with optimal fixed rotation (*Sekihara et al., 2004*). Significant brain areas and peak coordinates were related to brain areas using the Anatomical Automatic Labeling (AAL) atlas for SPM8 (*Tzourio-Mazoyer et al., 2002*). After statistical analysis, source localised data was interpolated onto the Colin 27 Average Brain Model MRI. Cerebellar and brainstem interpolations were excluded from the coordinate system.

## Statistical analysis

Condition differences in the behavioural task were estimated with paired *t*-tests and repeated measures ANOVAs, utilising Tukey–Kramer post hoc test. Power differences between conditions, as well as source-space contrasts were analysed using cluster permutation analysis (*Maris and Oostenveld, 2007*). In this procedure, condition labels were randomly shuffled 1000 times, creating pairs of surrogate conditions. To test for significance, paired *t*-tests were conducted for each data point and each channel, resulting in one *t*-matrix for real conditions and 1000 *t*-matrices for surrogate conditions. Significant *t*-values (p<0.05) were defined as clusters if there was at least one significant data point present at the same time and frequency in at least two neighbouring channels. To correct for multiple comparisons, a condition difference was only assumed if the maximum sum of *t*-values in a real cluster exceeded the same sum of 95% of the clusters found in the surrogate data. To replicate our results in the second study, we applied the same statistics and averaged the previously found time-intervals into windows of 500 ms (e.g. if we found an effect –0.51 to –0.0620 s prior to target onset we tested this effect for a time window of –0.5–0 s prior to target onset).

The relationship between induced changes in alpha activity and frequency-tagged responses was assessed using trial by trial Spearman correlations. For each participant and each electrode, a correlation coefficient was calculated between the average activity in a previously identified cluster, which was used as seed (e.g. condition differences in alpha activity), and the average activity of the electrophysiological correlate of interest (e.g. 36 Hz activity over the previously identified time window). The correlation coefficients were z-transformed, and the resulting channel by participant matrix was tested against null-correlation model using the cluster permutation approach described above. Derived clusters were additionally tested and visualised by comparing median split trials of high vs low activity. For this analysis, outliers (values deviating more than 2 standard deviations from the mean) were excluded. Furthermore, the average correlation coefficient of the cluster was tested against a 0-correlation model for each participant using *t*-statistics. Lastly, an interaction between electrophysiological correlations and conditions was performed using correlation coefficients for each participant and electrode, testing them between conditions using the cluster permutation approach. Perceptually

uniform and universally readable colormaps were applied to all visualisations (*Crameri et al., 2020*). All data are presented as mean ± standard error of the mean (SEM).

## Acknowledgements

This work was made possible by funding support from Facebook Oculus and BBSRC (BB/R018723/1).

## Additional information

### Funding

| Funder | Grant reference number | Author |
| --- | --- | --- |
| BBSRC | BB/R018723/1 | Ole Jensen |
| Facebook Oculus | | Ali Mazaheri |

The funders had no role in study design, data collection and interpretation, or the decision to submit the work for publication.

### Author contributions

Marion Brickwedde, Conceptualization, Data curation, Formal analysis, Investigation, Visualization, Methodology, Writing – original draft; Rupali Limachya, Emma Sutton, Data curation, Investigation, Writing – review and editing; Roksana Markiewicz, Data curation, Supervision, Validation, Investigation, Writing – review and editing; Christopher Postzich, Data curation, Formal analysis, Writing – review and editing; Kimron Shapiro, Supervision, Writing – review and editing; Ole Jensen, Supervision, Methodology, Writing – review and editing; Ali Mazaheri, Conceptualization, Resources, Software, Supervision, Funding acquisition, Validation, Project administration, Writing – review and editing

### Author ORCIDs

Marion Brickwedde (iD) https://orcid.org/0000-0002-3461-038X
Ole Jensen (iD) https://orcid.org/0000-0001-8193-8348

### Ethics

All participants provided written informed consent. Ethical approval for the study was obtained from the Ethics Committee of Birmingham University.

Reviewer #1 (Public review): https://doi.org/10.7554/eLife.106050.4.sa1
Author response https://doi.org/10.7554/eLife.106050.4.sa2

## Additional files

### Supplementary files

MDAR checklist

### Data availability

The EEG-dataset collected for this study has been deposited at: https://doi.org/10.18112/openneuro.ds007648.v1.1.0 and is publicly available. In the derivatives folder, data and code to replicate the main results (*Figure 4*) can be accessed.The MEG-dataset collected for this study has been deposited at: https://doi.org/10.18112/openneuro.ds007663.v1.0.0 and is publicly available. In the derivatives folder, data and code to replicate the main results (*Figure 9*) can be accessed.

The following datasets were generated:

| Author(s) | Year | Dataset title | Dataset URL | Database and Identifier |
|-----------|------|---------------|-------------|-------------------------|
| Brickwedde M, Limachya R, Markiewicz R, Sutton E, Postzich C, Shapiro K, Jensen O, Mazaheri A | 2026 | CrossModal Study | https://doi.org/10.18112/openneuro.ds007648.v1.1.0 | OpenNeuro, 10.18112/openneuro.ds007648.v1.1.0 |
| Brickwedde M, Limachya R, Markiewicz R, Sutton E, Postzich C, Shapiro K, Jensen O, Mazaheri A | 2026 | CrossModal Study | https://doi.org/10.18112/openneuro.ds007663.v1.0.0 | OpenNeuro, 10.18112/openneuro.ds007663.v1.0.0 |

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
