## [Editor Report · eLife Assessment]

This **valuable** manuscript provides **solid** evidence regarding the role of alpha oscillations in sensory gain control. The authors use an attention-cuing task in an initial EEG study followed by a separate MEG replication study to demonstrate that whilst (occipital) alpha oscillations are increased when anticipating an auditory target, so is visual responsiveness as assessed with frequency tagging. The findings offer a re-interpretation of the inhibitory role of the alpha rhythm, supporting that alpha oscillations contribute to interareal communication.

---

## [Referee Report · Reviewer #1 (Public review)]

In this study, Brickwedde et al. leveraged a cross-modal task where visual cues indicated whether upcoming targets required visual or auditory discrimination. Visual and auditory targets were paired with auditory and visual distractors, respectively. The authors found that during the cue-to-target interval, posterior alpha activity increased along with auditory and visual frequency-tagged activity when subjects were anticipating auditory targets. The authors conclude that their results imply that alpha modulation does not solely regulate 'gain control' in early visual areas (also referred to as alpha inhibition hypothesis), but rather orchestrates signal transmission to later stages of the processing stream.

Comments on the first revision:

I thank the authors for their clarifications. The manuscript is much improved now, in my opinion. The new power spectral density plots and revised Figure 1 are much appreciated. However, there is one remaining point that I am unclear about. In the rebuttal, the authors state the following: "To directly address the question of whether the auditory signal was distracting, we conducted a follow-up MEG experiment. In this study, we observed a significant reduction in visual accuracy during the second block when the distractor was present (see Fig. 7B and Suppl. Fig. 1B), providing clear evidence of a distractor cost under conditions where performance was not saturated."

I am very confused by this statement, because both Fig. 7B and Suppl. Fig. 1B show that the visual- (i.e., visual target presented alone) has a lower accuracy and longer reaction time than visual+ (i.e., visual target presented with distractor). In fact, Suppl. Fig. 1B legend states the following: "accuracy: auditory- - auditory+: M = 7.2 %; SD = 7.5; p = .001; t(25) = 4.9; visual- - visual+: M = -7.6%; SD = 10.80; p < .01; t(25) = -3.59; Reaction time: auditory- - auditory +: M = -20.64 ms; SD = 57.6; n.s.: p = .08; t(25) = -1.83; visual- - visual+: M = 60.1 ms ; SD = 58.52; p < .001; t(25) = 5.23."

These statements appear to directly contradict each other. I appreciate that the difficulty of auditory and visual trials in block 2 of MEG experiments are matched, but this does not address the question of whether the distractor was actually distracting (and thus needed to be inhibited by occipital alpha). Please clarify.

Comments on the latest version:

I am satisfied with the author's response and do not have any additional comments.

---

## [Author Response]

The following is the authors’ response to the current reviews.

I thank the authors for their clarifications. The manuscript is much improved now, in my opinion. The new power spectral density plots and revised Figure 1 are much appreciated. However, there is one remaining point that I am unclear about. In the rebuttal, the authors state the following: "To directly address the question of whether the auditory signal was distracting, we conducted a follow-up MEG experiment. In this study, we observed a significant reduction in visual accuracy during the second block when the distractor was present (see Fig. 7B and Suppl. Fig. 1B), providing clear evidence of a distractor cost under conditions where performance was not saturated."I am very confused by this statement, because both Fig. 7B and Suppl. Fig. 1B show that the visual- (i.e., visual target presented alone) has a lower accuracy and longer reaction time than visual+ (i.e., visual target presented with distractor). In fact, Suppl. Fig. 1B legend states the following: "accuracy: auditory- - auditory+: M = 7.2 %; SD = 7.5; p = .001; t(25) = 4.9; visual- - visual+: M = -7.6%; SD = 10.80; p < .01; t(25) = -3.59; Reaction time: auditory- - auditory +: M = -20.64 ms; SD = 57.6; n.s.: p = .08; t(25) = -1.83; visual- - visual+: M = 60.1 ms ; SD = 58.52; p < .001; t(25) = 5.23."These statements appear to directly contradict each other. I appreciate that the difficulty of auditory and visual trials in block 2 of MEG experiments are matched, but this does not address the question of whether the distractor was actually distracting (and thus needed to be inhibited by occipital alpha). Please clarify.

We apologize for mixing up the visual and auditory distractor cost in our rebuttal. The reviewer is right in that our two statements contradict each other.

To clarify: In the EEG experiment, we see significant distractor cost for auditory distractors in the accuracy (which can be seen in SUPPL Fig. 1A). We also see a faster reaction time with auditory distractors, which may speak to intersensory facilitation. As we used the same distractors for both experiments, it can be assumed that they were distracting in both experiments.

In our follow-up MEG-experiment, as the reviewer stated, performance in block 2 was higher than in block 1, even though there were distractors present. In this experiment, distractor cost and learning effects are difficult to disentangle. It is possible that participants improved over time for the visual discrimination task in Block 1, as performance at the beginning was quite low. To illustrate this, we divided the trials of each condition into bins of 10 and plotted the mean accuracy in these bins over time (see Author response image 1). Here it can be seen that in Block 2, there is a more or less stable performance over time with a variation < 10 %. In Block 1, both for visual as well as auditory trials, an improvement over time can be seen. This is especially strong for visual trials, which span a difference of > 20%. Note that the mean performance for the 80-90 trial bin was higher than any mean performance observed in Block 2.

Additionally, the same paradigm has been applied in previous investigations, which also found distractor costs for the here-used auditory stimuli in blocked and non-blocked designs. See:

Mazaheri, A., van Schouwenburg, M. R., Dimitrijevic, A., Denys, D., Cools, R., & Jensen, O. (2014). Region-specific modulations in oscillatory alpha activity serve to facilitate processing in the visual and auditory modalities. NeuroImage, 87, 356–362. https://doi.org/10.1016/j.neuroimage.2013.10.052

Van Diepen, R & Mazaheri, A 2017, 'Cross-sensory modulation of alpha oscillatory activity: suppression, idling and default resource allocation', European Journal of Neuroscience, vol. 45, no. 11, pp. 1431-1438. https://doi.org/10.1111/ejn.13570

**Author response image 1. sa2fig1:** Accuracy development over time in the MEG experiment. During block 1, a performance increase over time can be observed for visual as well as for auditory stimuli. During Block 2, performance is stable over time. Data are presented as mean ± SEM. N = 27 (one participant was excluded from this analysis, as their trial count in at least one condition was below 90 trials).

The following is the authors’ response to the previous reviews

**Reviewer #1 (Public review):**
In this study, Brickwedde et al. leveraged a cross-modal task where visual cues indicated whether upcoming targets required visual or auditory discrimination. Visual and auditory targets were paired with auditory and visual distractors, respectively. The authors found that during the cue-to-target interval, posterior alpha activity increased along with auditory and visual frequency-tagged activity when subjects were anticipating auditory targets. The authors conclude that their results disprove the alpha inhibition hypothesis, and instead implies that alpha "regulates downstream information transfer." However, as I detail below, I do not think the presented data irrefutably disproves the alpha inhibition hypothesis. Moreover, the evidence for the alternative hypothesis of alpha as an orchestrator for downstream signal transmission is weak. Their data serves to refute only the most extreme and physiologically implausible version of the alpha inhibition hypothesis, which assumes that alpha completely disengages the entire brain area, inhibiting all neuronal activity.

We thank the reviewer for taking the time to provide additional feedback and suggestions and we improved our manuscript accordingly.

(1) Authors assign specific meanings to specific frequencies (8-12 Hz alpha, 4 Hz intermodulation frequency, 36 Hz visual tagging activity, 40 Hz auditory tagging activity), but the results show that spectral power increases in all of these frequencies towards the end of the cue-to-target interval. This result is consistent with a broadband increase, which could simply be due to additional attention required when anticipating auditory target (since behavioral performance was lower with auditory targets, we can say auditory discrimination was more difficult). To rule this out, authors will need to show a power spectral density curve with specific increases around each frequency band of interest. In addition, it would be more convincing if there was a bump in the alpha band, and distinct bumps for 4 vs 36 vs 40 Hz band.

This is an interesting point with several aspects, which we will address separately

Broadband Increase vs. Frequency-Specific Effects:

The suggestion that the observed spectral power increases may reflect a broadband effect rather than frequency-specific tagging is important. However, Supplementary Figure 11 shows no difference between expecting an auditory or visual target at 44 Hz. This demonstrates that (1) there is no uniform increase across all frequencies, and (2) the separation between our stimulation frequencies was sufficient to allow differentiation using our method.

Task Difficulty and Performance Differences:

The reviewer suggests that the observed effects may be due to differences in task difficulty, citing lower performance when anticipating auditory targets in the EEG study. This issue was explicitly addressed in our follow-up MEG study, where stimulus difficulty was calibrated. In the second block—used for analysis—accuracy between auditory and visual targets was matched (see Fig. 7B). The replication of our findings under these controlled conditions directly rules out task difficulty as the sole explanation. This point is clearly presented in the manuscript.

Power Spectrum Analysis:

The reviewer’s suggestion that our analysis lacks evidence of frequency-specific effects is addressed directly in the manuscript. While we initially used the Hilbert method to track the time course of power fluctuations, we also included spectral analyses to confirm distinct peaks at the stimulation frequencies. Specifically, when averaging over the alpha cluster, we observed a significant difference at 10 Hz between auditory and visual target expectation, with no significant differences at 36 or 40 Hz in that cluster. Conversely, in the sensor cluster showing significant 36 Hz activity, alpha power did not differ, but both 36 Hz and 40 Hz tagging frequencies showed significant effects These findings clearly demonstrate frequency-specific modulation and are already presented in the manuscript.

(2) For visual target discrimination, behavioral performance with and without the distractor is not statistically different. Moreover, the reaction time is faster with distractor. Is there any evidence that the added auditory signal was actually distracting?

We appreciate the reviewer’s observation regarding the lack of a statistically significant difference in behavioral performance for visual target discrimination with and without the auditory distractor. While this was indeed the case in our EEG experiment, we believe the absence of an accuracy effect may be attributable to a ceiling effect, as overall visual performance approached 100%. This high baseline likely masked any subtle influence of the distractor.

To directly address the question of whether the auditory signal was distracting, we conducted a follow-up MEG experiment. In this study, we observed a significant reduction in visual accuracy during the second block when the distractor was present (see Fig. 7B and Suppl. Fig. 1B), providing clear evidence of a distractor cost under conditions where performance was not saturated.

Regarding the faster reaction times observed in the presence of the auditory distractor, this phenomenon is consistent with prior findings on intersensory facilitation. Auditory stimuli, which are processed more rapidly than visual stimuli, can enhance response speed to visual targets—even when the auditory input is non-informative or nominally distracting (Nickerson, 1973; Diederich & Colonius, 2008; Salagovic & Leonard, 2021). Thus, while the auditory signal may facilitate motor responses, it can simultaneously impair perceptual accuracy, depending on task demands and baseline performance levels.

Taken together, our data suggest that the auditory signal does exert a distracting influence, particularly under conditions where visual performance is not at ceiling. The dual effect—facilitated reaction time but reduced accuracy—highlights the complexity of multisensory interactions and underscores the importance of considering both behavioral and neurophysiological measures.

(3) It is possible that alpha does suppress task-irrelevant stimuli, but only when it is distracting. In other words, perhaps alpha only suppresses distractors that are presented simultaneously with the target. Since the authors did not test this, they cannot irrefutably reject the alpha inhibition hypothesis.

The reviewer’s claim that we did not test whether alpha suppresses distractors presented simultaneously with the target is incorrect. As stated in the manuscript and supported by our data (see point 2), auditory distractors were indeed presented concurrently with visual targets, and they were demonstrably distracting. Therefore, the scenario the reviewer suggests was not only tested—it forms a core part of our design.

Furthermore, it was never our intention to irrefutably reject the alpha inhibition hypothesis. Rather, our aim was to revise and expand it. If our phrasing implied otherwise, we have now clarified this in the manuscript. Specifically, we propose that alpha oscillations:

(a) Exhibit cyclic inhibitory and excitatory dynamics;

(b) Regulate processing by modulating transfer pathways, which can result in either inhibition or facilitation depending on the network context.

In our study, we did not observe suppression of distractor transfer, likely due to the engagement of a supramodal system that enhances both auditory and visual excitability. This interpretation is supported by prior findings (e.g., Jacoby et al., 2012), which show increased visual SSEPs under auditory task load, and by Zhigalov et al. (2020), who found no trial-by-trial correlation between alpha power and visual tagging in early visual areas, despite a general association with attention.

Recent evidence (Clausner et al., 2024; Yang et al., 2024) further supports the notion that alpha oscillations serve multiple functional roles depending on the network involved. These roles include intra- and inter-cortical signal transmission, distractor inhibition, and enhancement of downstream processing (Scheeringa et al., 2012; Bastos et al., 2015; Zumer et al., 2014). We believe the most plausible account is that alpha oscillations support both functions, depending on context.

To reflect this more clearly, we have updated Figure 1 to present a broader signal-transfer framework for alpha oscillations, beyond the specific scenario tested in this study.

We have now revised Figure 1 and several sentences in the introduction and discussion, to clarify this argument.

L35-37: Previous research gave rise to the prominent alpha inhibition hypothesis, which suggests that oscillatory activity in the alpha range (~10 Hz) plays a mechanistic role in selective attention through functional inhibition of irrelevant cortical areas (see Fig. 1; Foxe et al., 1998; Jensen & Mazaheri, 2010; Klimesch et al., 2007).

L60-65: In contrast, we propose that functional and inhibitory effects of alpha modulation, such as distractor inhibition, are exhibited through blocking or facilitating signal transmission to higher order areas (Peylo et al., 2021; Yang et al., 2023; Zhigalov & Jensen, 2020; Zumer et al., 2014), gating feedforward or feedback communication between sensory areas (see Fig. 1; Bauer et al., 2020; Haegens et al., 2015; Uemura et al., 2021).

L482-485: This suggests that responsiveness of the visual stream was not inhibited when attention was directed to auditory processing and was not inhibited by occipital alpha activity, which directly contradicts the proposed mechanism behind the alpha inhibition hypothesis.

L517-519: Top-down cued changes in alpha power have now been widely viewed to play a functional role in directing attention: the processing of irrelevant information is attenuated by increasing alpha power in areas involved with processing this information (Foxe, Simpson, & Ahlfors, 1998; Hanslmayr et al., 2007; Jensen & Mazaheri, 2010).

L566-569: As such, it is conceivable that alpha oscillations can in some cases inhibit local transmission, while in other cases, depending on network location, connectivity and demand, alpha oscillation can facilitate signal transmission. This mechanism allows to increase transmission of relevant information and to block transmission of distractors.

(4) In the abstract and Figure 1, the authors claim an alternative function for alpha oscillations; that alpha "orchestrates signal transmission to later stages of the processing stream." In support, the authors cite their result showing that increased alpha activity originating from early visual cortex is related to enhanced visual processing in higher visual areas and association areas. This does not constitute a strong support for the alternative hypothesis. The correlation between posterior alpha power and frequency-tagged activity was not specific in any way; Fig. 10 shows that the correlation appeared on both (1) anticipating-auditory and anticipating-visual trials, (2) the visual tagged frequency and the auditory tagged activity, and (3) was not specific to the visual processing stream. Thus, the data is more parsimonious with a correlation than a causal relationship between posterior alpha and visual processing.

Again, the reviewer raises important points, which we want to address

The correlation between posterior alpha power and frequency-tagged activity was not specific, as it is present both when auditory and visual targets are expected:

If there is a connection between posterior alpha activity and higher-order visual information transfer, then it can be expected that this relationship remains across conditions and that a higher alpha activity is accompanied by higher frequency-tagged activity, both over trials and over conditions. However, it is possible that when alpha activity is lower, such as when expecting a visual target, the signal-to-noise ratio is affected, which may lead to higher difficulty to find a correlation effect in the data when using non-invasive measurements.

The connection between alpha activity and frequency-tagged activity appears both for auditory as well as visual stimuli and The correlation is not specific to the visual processing stream:

While we do see differences between conditions (e.g. in the EEG-analysis, mostly 36 Hz correlated with alpha activity and only in one condition 40 Hz showed a correlation as well), it is true that in our MEG analysis, we found correlations both between alpha activity and 36 Hz as well as alpha activity and 40 Hz.

We acknowledge that when analysing frequency-tagged activity on a trial-by-trial basis, where removal of non-timelocked activity through averaging (which we did when we tested for condition differences in Fig. 4 and 9) is not possible, there is uncertainty in the data. Baseline-correction can alleviate this issue, but it cannot offset the possibility of non-specific effects. We therefore decided to repeat the analysis with a fast-fourier calculated power instead of the Hilbert power, in favour of a higher and stricter frequency-resolution, as we averaged over a time-period and thus, the time-domain was not relevant for this analysis. In this more conservative analysis, we can see that only 36 Hz tagged activity when expecting an auditory target correlated with early visual alpha activity.

Additionally, we added correlation analyses between alpha activity and frequency-tagged activity within early visual areas, using the sensor cluster which showed significant condition differences in alpha activity. Here, no correlations between frequency-tagged activity and alpha activity could be found (apart from a small correlation with 40 Hz which could not be confirmed by a median split; see SUPPL Fig. 14 C). The absence of a significant correlation between early visual alpha and frequency-tagged activity has previously been described by others (Zhigalov & Jensen, 2020) and a Bayes factor of below 1 also indicated that the alternative hypotheses is unlikely.

Nonetheless, a correlation with auditory signal is possible and could be explained in different ways. For example, it could be that very early auditory feedback in early visual cortex (see for example Brang et al., 2022) is transmitted alongside visual information to higher-order areas. Several studies have shown that alpha activity and visual as well as auditory processing are closely linked together (Bauer et al., 2020; Popov et al., 2023). Inference on whether or how this link could play out in the case of this manuscript expands beyond the scope of this study.

To summarize, we believe the fact that 36 Hz activity within early visual areas does not correlate with alpha activity on a trial-by-trial basis, but that 36 Hz activity in other areas does, provides strong evidence that alpha activity affects down-stream signal processing.

We mention this analysis now in our discussion:

L533-536: Our data provides evidence in favour of this view, as we can show that early sensory alpha activity does not covary over trials with SSEP magnitude in early visual areas, but covaries instead over trials with SSEP magnitude in higher order sensory areas (see also SUPPL. Fig. 14).

**Reviewer #1 (Recommendations for the authors):**
The evidence for the alternative hypothesis, that alpha in early sensory areas orchestrates downstream signal transmission, is not strong enough to be described up front in the abstract and Figure 1. I would leave it in the Discussion section, but advise against mentioning it in the abstract and Figure 1.

We appreciate the reviewer’s concern regarding the inclusion of the alternative hypothesis—that alpha activity in early sensory areas orchestrates downstream signal transmission—in the abstract and Figure 1. While we agree that this interpretation is still developing, recent studies (Keitel et al., 2025; Clausner et al., 2024; Yang et al., 2024) provide growing support for this framework.

In response, we have revised the introduction, discussion, and Figure 1 to clarify that our intention is not to outright dismiss the alpha inhibition hypothesis, but to refine and expand it in light of new data. This revision does not invalidate the prior literature on alpha timing and inhibition; rather, it proposes an updated mechanism that may better account for observed effects.

We have though retained Figure 1, as it visually contextualizes the broader theoretical landscape. while at the same time added further analyses to strengthen our empirical support for this emerging view.

References:

Bastos, A. M., Litvak, V., Moran, R., Bosman, C. A., Fries, P., & Friston, K. J. (2015). A DCM study of spectral asymmetries in feedforward and feedback connections between visual areas V1 and V4 in the monkey. NeuroImage, 108, 460–475. https://doi.org/10.1016/j.neuroimage.2014.12.081

Bauer, A. R., Debener, S., & Nobre, A. C. (2020). Synchronisation of Neural Oscillations and Cross-modal Influences. Trends in cognitive sciences, 24(6), 481–495. https://doi.org/10.1016/j.tics.2020.03.003

Brang, D., Plass, J., Sherman, A., Stacey, W. C., Wasade, V. S., Grabowecky, M., Ahn, E., Towle, V. L., Tao, J. X., Wu, S., Issa, N. P., & Suzuki, S. (2022). Visual cortex responds to sound onset and offset during passive listening. Journal of neurophysiology, 127(6), 1547–1563. https://doi.org/10.1152/jn.00164.2021

Clausner T., Marques J., Scheeringa R. & Bonnefond M (2024). Feature specific neuronal oscillations in cortical layers BioRxiv :2024.07.31.605816. https://doi.org/10.1101/2024.07.31.605816

Diederich, A., & Colonius, H. (2008). When a high-intensity "distractor" is better then a low-intensity one: modeling the effect of an auditory or tactile nontarget stimulus on visual saccadic reaction time. Brain research, 1242, 219–230. https://doi.org/10.1016/j.brainres.2008.05.081

Haegens, S., Nácher, V., Luna, R., Romo, R., & Jensen, O. (2011). α-Oscillations in the monkey sensorimotor network influence discrimination performance by rhythmical inhibition of neuronal spiking. Proceedings of the National Academy of Sciences of the United States of America, 108(48), 19377–19382. https://doi.org/10.1073/pnas.1117190108

Jacoby, O., Hall, S. E., & Mattingley, J. B. (2012). A crossmodal crossover: opposite effects of visual and auditory perceptual load on steady-state evoked potentials to irrelevant visual stimuli. NeuroImage, 61(4), 1050–1058. https://doi.org/10.1016/j.neuroimage.2012.03.040

Keitel, A., Keitel, C., Alavash, M., Bakardjian, K., Benwell, C. S. Y., Bouton, S., Busch, N. A., Criscuolo, A., Doelling, K. B., Dugue, L., Grabot, L., Gross, J., Hanslmayr, S., Klatt, L.-I., Kluger, D. S., Learmonth, G., London, R. E., Lubinus, C., Martin, A. E., … Kotz, S. A. (2025). Brain rhythms in cognition – controversies and future directions. ArXiv. https://doi.org/10.48550/arXiv.2507.15639

Nickerson R. S. (1973). Intersensory facilitation of reaction time: energy summation or preparation enhancement?. Psychological review, 80(6), 489–509. https://doi.org/10.1037/h0035437

Popov, T., Gips, B., Weisz, N., & Jensen, O. (2023). Brain areas associated with visual spatial attention display topographic organization during auditory spatial attention. Cerebral cortex (New York, N.Y. : 1991), 33(7), 3478–3489. https://doi.org/10.1093/cercor/bhac285

Salagovic, C. A., & Leonard, C. J. (2021). A nonspatial sound modulates processing of visual distractors in a flanker task. Attention, perception & psychophysics, 83(2), 800–809. https://doi.org/10.3758/s13414-020-02161-5

Scheeringa, R., Petersson, K. M., Kleinschmidt, A., Jensen, O., & Bastiaansen, M. C. (2012). EEG α power modulation of fMRI resting-state connectivity. Brain connectivity, 2(5), 254–264. https://doi.org/10.1089/brain.2012.0088

Spaak, E., Bonnefond, M., Maier, A., Leopold, D. A., & Jensen, O. (2012). Layer-specific entrainment of γ-band neural activity by the α rhythm in monkey visual cortex. Current biology : CB, 22(24), 2313–2318. https://doi.org/10.1016/j.cub.2012.10.020

Yang, X., Fiebelkorn, I. C., Jensen, O., Knight, R. T., & Kastner, S. (2024). Differential neural mechanisms underlie cortical gating of visual spatial attention mediated by alpha-band oscillations. Proceedings of the National Academy of Sciences of the United States of America, 121(45), e2313304121. https://doi.org/10.1073/pnas.2313304121

Zhigalov, A., & Jensen, O. (2020). Alpha oscillations do not implement gain control in early visual cortex but rather gating in parieto-occipital regions. Human brain mapping, 41(18), 5176–5186. https://doi.org/10.1002/hbm.25183

Zumer, J. M., Scheeringa, R., Schoffelen, J. M., Norris, D. G., & Jensen, O. (2014). Occipital alpha activity during stimulus processing gates the information flow to object-selective cortex. PLoS biology, 12(10), e1001965. https://doi.org/10.1371/journal.pbio.1001965